# The Implicit Bias of AdaGrad on Separable Data

**Qian Qian**
Department of Statistics
Ohio State University
Columbus, OH 43210, USA
qian.216@osu.edu

**Xiaoyuan Qian**
School of Mathematical Sciences
Dalian University of Technology
Dalian, Liaoning 116024, China
xyqian@dlut.edu.cn

## Abstract

We study the implicit bias of AdaGrad on separable linear classification problems. We show that AdaGrad converges to a direction that can be characterized as the solution of a quadratic optimization problem with the same feasible set as the hard SVM problem. We also give a discussion about how different choices of the hyperparameters of AdaGrad might impact this direction. This provides a deeper understanding of why adaptive methods do not seem to have the generalization ability as good as gradient descent does in practice.

## 1 Introduction

In recent years, implicit regularization from various optimization algorithms plays a crucial role in the generalization abilities in training deep neural networks (Salakhutdinov and Srebro [2015], Neyshabur et al. [2015], Keskar et al. [2016], Neyshabur et al. [2017], Zhang et al. [2017]). For example, in underdetermined problems where the number of parameters is larger than the number of training examples, many global optima fail to exhibit good generalization properties, however, a specific optimization algorithm (such as gradient descent) does converge to a particular optimum that generalize well, although no explicit regularization is enforced when training the model. In other words, the optimization technique itself "biases" towards a certain model in an implicit way (Soudry et al. [2018]). This motivates a line of works to investigate the implicit biases of various algorithms (Telgarsky [2013], Soudry et al. [2018], Gunasekar et al. [2017, 2018a,b]).

The choice of algorithms would affect the implicit regularization introduced in the learned models. In underdetermined least squares problems, where the minimizers are finite, we know that gradient descent yields the minimum $L_2$ norm solution, whereas coordinate descent might give a different solution. Another example is logistic regression with separable data. While gradient descent converges in the direction of the hard margin support vector machine solution (Soudry et al. [2018]), coordinate descent converges to the maximum $L_1$ margin solution (Telgarsky [2013], Gunasekar et al. [2018a]). Unlike the squared loss, the logistic loss does not admit a finite global minimizer on separable data: the iterates will diverge in order to drive the loss to zero. As a result, instead of characterizing the convergence of the iterates $\boldsymbol{w}(t)$, it is the asymptotic direction of these iterates i.e., $\lim_{t \to \infty} \boldsymbol{w}(t)/\|\boldsymbol{w}(t)\|$ that is important and therefore has been characterized (Soudry et al. [2018], Gunasekar et al. [2018b]).

Morevoer, it has attracted much attention that different adaptive methods of gradient descent and hyperparameters of an adaptive method exhibit different biases, thus leading to different generalization performance in deep learning (Salakhutdinov and Srebro [2015], Keskar et al. [2016], Wilson et al. [2017], Hoffer et al. [2017]). Among those findings is that the vanilla SGD algorithm demonstrates better generalization than its adaptive variants (Wilson et al. [2017]), such as AdaGrad (Duchi et al. [2010]) and Adam (Kingma and Ba [2015]). Therefore it is important to precisely characterize how different adaptive methods induce difference biases. A natural question to ask is: can we explain this observation by characterizing the implicit bias of AdaGrad, which is a paradigm of adaptive

methods, in a binary classification setting with separable data using logistic regression? And how does the implicit bias depend on the choice of the hyperparameters of this specific algorithm, such as initialization, step sizes, etc?

## 1.1 Our Contribution

In this work we study Adagrad applied to logistic regression with separable data. Our contribution is three-fold as listed as follows.

- We prove that the directions of AdaGrad iterates, with a constant step size sufficiently small, always converge.
- We formulate the asymptotic direction as the solution of a quadratic optimization problem. This achieves a theoretical characterization of the implicit bias of AdaGrad, which also provides insights about why and how the factors involved, such as certain intrinsic properties of the dataset, the initialization and the learning rate, affect the implicit bias.
- We introduce a novel approach to study the bias of AdaGrad. It is mainly based on a geometric estimation on the directions of the updates, which doesn't depend on any calculation on the convergence rate.

## 1.2 Paper Organization

This paper is organized as follows. In Section 2 we explain our problem setup. The main theory is developed in Section 3, including convergence of the adaptive learning rates of AdaGrad, existence of the asymptotic direction of AdaGrad iterates, and relations between the asymptotic directions of Adagrad and gradient descent iterates. We conclude our paper in Section 4 with a review of our results and some questions left to future research.

## 2 Problem Setup

Let $\{(\boldsymbol{x}_n, y_n): \ n = 1, \cdots, N\}$ be a training dataset with features $\boldsymbol{x}_n \in \mathbb{R}^p$ and labels $y_n \in \{-1, 1\}$. To simplify the notation, we redefine $y_n \boldsymbol{x}_n$ as $\boldsymbol{x}_n$, $n = 1, \cdots, N$, and consider learning the logistic regression model over the empirical loss:

$$\mathcal{L}(\boldsymbol{w}) = \sum_{n=1}^{N} l\left(\boldsymbol{w}^T \boldsymbol{x}_n\right), \quad \boldsymbol{w} \in \mathbb{R}^p,$$

where $l: \mathbb{R}^p \to \mathbb{R}$. We focus on the following case, same as proposed in Soudry et al. [2018]:

**Assumption 1.** There exists a vector $\boldsymbol{w}_*$ such that $\boldsymbol{w}_*^T \boldsymbol{x}_n > 0$ for all $n$.

**Assumption 2.** $l(u)$ is continuously differentiable, $\beta-$smooth, and strictly decreasing to zero.

**Assumption 3.** There exist positive constants $a, b, c$, and $d$ such that

$$\left| l'(u) + ce^{-au} \right| \le e^{-(a+b)u}, \ \text{ for } u > d.$$

It is easy to see that the exponential loss $l(u) = e^{-u}$ and the logistic loss $l(u) = \log\left(1 + e^{-u}\right)$ both meet these assumptions.

Given two hyperparameters $\epsilon$, $\eta > 0$ and an initial point $\boldsymbol{w}(0) \in \mathbb{R}^p$, we consider the diagonal AdaGrad iterates

$$\boldsymbol{w}(t+1) = \boldsymbol{w}(t) - \eta \boldsymbol{h}(t) \odot \boldsymbol{g}(t), \quad t = 0, 1, 2, \cdots, \tag{1}$$

where

$$\boldsymbol{g}(t) = (g_1(t), \cdots, g_p(t)),$$

$$g_i(t) = \frac{\partial \mathcal{L}}{\partial w_i}(\boldsymbol{w}(t)),$$

$$\boldsymbol{h}(t) = (h_1(t), \cdots, h_p(t)),$$

$$h_i(t) = \frac{1}{\sqrt{g_i(0)^2 + \cdots + g_i(t)^2 + \epsilon}}, \quad i = 1, \cdots, p,$$

and $\odot$ is the element-wise multiplication of two vectors, e.g.

$$\boldsymbol{a} \odot \boldsymbol{b} = (a_1 b_1, \cdots, a_p b_p)^T$$

for $\boldsymbol{a} = (a_1, \cdots, a_p)^T$, $\boldsymbol{b} = (b_1, \cdots, b_p)^T$.

To analyze the convergence of the algorithm, we put an additional restriction on the hyperparameter $\eta$.

**Assumption 4.** The hyperparameter $\eta$ is not too large; specifically,

$$\eta < \frac{2 \, \min_{i \in \{1, \cdots, p\}} \sqrt{g_i(0)^2 + \epsilon}}{\beta}. \tag{2}$$

We are interested in the asymptotic behavior of the AdaGrad iteration scheme in (1). The main problem is: does there exists some vector $\boldsymbol{w}_A$ such that

$$\lim_{t \to \infty} \boldsymbol{w}(t) / \|\boldsymbol{w}(t)\| = \boldsymbol{w}_A ?$$

We will provide an affirmative answer to this question in the following section.

## 3 The Asymptotic Direction of AdaGrad Iterates

### 3.1 Convergence of the Adaptive Learning Rates

We first provide some elementary facts about AdaGrad iterates (1) with all assumptions (1-4) proposed in Section 2.

**Lemma 3.1.** $\mathcal{L}(\boldsymbol{w}(t+1)) < \mathcal{L}(\boldsymbol{w}(t))$ ($t = 0, 1, \cdots$).

**Lemma 3.2.** $\sum_{t=0}^{\infty} \|\boldsymbol{g}(t)\|^2 < \infty$.

We notice that Gunasekar et al. [2018a] showed a similar result (Lemma 6, in Section 3.3 of their work) for exponential loss only, under slightly different assumptions. However, their approach depends on some specific properties of the exponential function, and thus cannot be extended to Lemma 3.2 in a trivial manner.

**Lemma 3.3.** The following statements hold:

(i) $\|\boldsymbol{g}(t)\| \to 0$ $(t \to \infty)$.

(ii) $\|\boldsymbol{w}(t)\| \to \infty$ $(t \to \infty)$.

(iii) $\mathcal{L}(\boldsymbol{w}(t)) \to 0$ $(t \to \infty)$.

(iv) $\forall n$, $\lim_{t \to \infty} \boldsymbol{w}(t)^T \boldsymbol{x}_n = \infty$.

(v) $\exists t_0$, $\forall t > t_0$, $\boldsymbol{w}(t)^T \boldsymbol{x}_n > 0$.

**Theorem 3.1.** The sequence $\{\boldsymbol{h}(t)\}_{t=0}^{\infty}$ converges as $t \to \infty$ to a vector

$$\boldsymbol{h}_{\infty} = (h_{\infty,1}, \cdots, h_{\infty,p})$$

satisfying $h_{\infty,i} > 0$ $(i = 1, \cdots, p)$.

### 3.2 Convergence of the Directions of AdaGrad Iterates

In the remainder of the paper we denote $\boldsymbol{h}_{\infty} = \lim_{t \to \infty} \boldsymbol{h}(t)$ and $\boldsymbol{\xi}_n = \boldsymbol{h}_{\infty}^{1/2} \odot \boldsymbol{x}_n$ $(n = 1, \cdots, N)$. Since, by Theorem 3.1, the components of $\boldsymbol{h}_{\infty}$ have a positive lower bound, we can define

$$\boldsymbol{\beta}(t) = \boldsymbol{h}_{\infty}^{-1} \odot \boldsymbol{h}(t) \ \ (t = 0, 1, 2, \cdots).$$

Here the squared root and the inverse of vectors are defined element-wise. We call the function

$$\mathcal{L}_{ind} : \mathbb{R}^p \to \mathbb{R}, \ \ \mathcal{L}_{ind}(\boldsymbol{v}) = \sum_{n=1}^{N} l\left(\boldsymbol{v}^T \boldsymbol{\xi}_n\right)$$

the *induced loss* with respect to AdaGrad (1). Note that

$$\mathcal{L}(\boldsymbol{w}) = \sum_{n=1}^{N} l\left(\boldsymbol{w}^T \boldsymbol{x}_n\right) = \sum_{n=1}^{N} l\left(\left(\boldsymbol{h}_\infty^{-1/2} \odot \boldsymbol{w}\right)^T \left(\boldsymbol{h}_\infty^{1/2} \odot \boldsymbol{x}_n\right)\right)$$

$$= \sum_{n=1}^{N} l\left(\left(\boldsymbol{h}_\infty^{-1/2} \odot \boldsymbol{w}\right)^T \boldsymbol{\xi}_n\right) = \mathcal{L}_{ind}\left(\boldsymbol{h}_\infty^{-1/2} \odot \boldsymbol{w}\right).$$

Thus if we set

$$\boldsymbol{v}(t) = \boldsymbol{h}_\infty^{-1/2} \odot \boldsymbol{w}(t) \quad (t = 0, 1, 2, \cdots), \tag{3}$$

then $\boldsymbol{v}(0) = \boldsymbol{h}_\infty^{-1/2} \odot \boldsymbol{w}(0)$, and

$$\boldsymbol{h}_\infty^{1/2} \odot \boldsymbol{v}(t+1) = \boldsymbol{w}(t+1) = \boldsymbol{w}(t) - \eta \boldsymbol{h}(t) \odot \nabla \mathcal{L}\left(\boldsymbol{w}(t)\right)$$

$$= \boldsymbol{h}_\infty^{1/2} \odot \boldsymbol{v}(t) - \eta \boldsymbol{h}(t) \odot \boldsymbol{h}_\infty^{-1/2} \odot \nabla \mathcal{L}\left(\boldsymbol{h}_\infty^{1/2} \odot \boldsymbol{v}(t)\right)$$

$$= \boldsymbol{h}_\infty^{1/2} \odot \boldsymbol{v}(t) - \eta \boldsymbol{h}(t) \odot \boldsymbol{h}_\infty^{-1/2} \odot \nabla \mathcal{L}_{ind}\left(\boldsymbol{h}_\infty^{-1/2} \odot \left(\boldsymbol{h}_\infty^{1/2} \odot \boldsymbol{v}(t)\right)\right)$$

$$= \boldsymbol{h}_\infty^{1/2} \odot \left(\boldsymbol{v}(t) - \eta \boldsymbol{\beta}(t) \odot \nabla \mathcal{L}_{ind}\left(\boldsymbol{v}(t)\right)\right),$$

or

$$\boldsymbol{v}(t+1) = \boldsymbol{v}(t) - \eta \boldsymbol{\beta}(t) \odot \nabla \mathcal{L}_{ind}(\boldsymbol{v}(t)) \quad (t = 0, 1, \cdots). \tag{4}$$

We refer to (4) as the *induced form* of AdaGrad (1).

The following result for the induced form is a simple corollary of Lemma 3.3.

**Lemma 3.4.** The following statements hold:

(i) $\|\nabla \mathcal{L}_{ind}(t)\| \to 0 \quad (t \to \infty)$.

(ii) $\|\boldsymbol{v}(t)\| \to \infty \quad (t \to \infty)$.

(iii) $\mathcal{L}_{ind}(\boldsymbol{v}(t)) \to 0 \quad (t \to \infty)$.

(iv) $\forall n, \quad \lim_{t \to \infty} \boldsymbol{v}(t)^T \boldsymbol{\xi}_n = \infty$.

(v) $\exists t_0, \quad \forall t > t_0, \quad \boldsymbol{v}(t)^T \boldsymbol{\xi}_n > 0$.

For the induced loss $\mathcal{L}_{ind}$, consider GD iterates

$$\boldsymbol{u}(t+1) = \boldsymbol{u}(t) - \eta \nabla \mathcal{L}_{ind}(\boldsymbol{u}(t)) \quad (t = 0, 1, \cdots). \tag{5}$$

According to Theorem 3 in Soudry et.al.(2018), we have

$$\lim_{t \to \infty} \frac{\boldsymbol{u}(t)}{\|\boldsymbol{u}(t)\|} = \frac{\widehat{\boldsymbol{u}}}{\|\widehat{\boldsymbol{u}}\|},$$

where

$$\widehat{\boldsymbol{u}} = \underset{\boldsymbol{u}^T \boldsymbol{\xi}_n \geq 1, \forall n}{\arg \min} \|\boldsymbol{u}\|^2.$$

Noting that $\boldsymbol{\beta}(t) \to \boldsymbol{1} \quad (t \to \infty)$ we can obtain GD iterates (5) by taking the limit of $\boldsymbol{\beta}(t)$ in (4). Therefore it is reasonable to expect that these two iterative processes have similar asymptotic behaviors, especially a common limiting direction.

Different from the case of GD method discussed in Soudry et al. [2018], however, it is difficult to obtain an effective estimation about the convergence rate of $\boldsymbol{w}(t)$. Instead, we introduce an orthogonal decomposition approach to obtain the asymptotic direction of the original Adagrad process (1).

In the remainder of the paper, we denote by $P$ the projection onto the $1-$dimensional subspace spanned by $\widehat{\boldsymbol{u}}$, and $Q$ the projection onto the orthogonal complement. Without any loss of generality we may assume $\|\widehat{\boldsymbol{u}}\| = 1$. Thus we have the orthogonal decomposition

$$\boldsymbol{v} = P\boldsymbol{v} + Q\boldsymbol{v} \quad (\boldsymbol{v} \in \mathbb{R}^p),$$

where $P\boldsymbol{v} = \|P\boldsymbol{v}\|\widehat{\boldsymbol{u}} = \left(\boldsymbol{v}^T \widehat{\boldsymbol{u}}\right) \widehat{\boldsymbol{u}}$. Moreover, we denote

$$\boldsymbol{\delta}(t) = -\eta \nabla \mathcal{L}_{ind}\left(\boldsymbol{v}(t)\right), \quad \boldsymbol{d}(t) = \boldsymbol{\beta}(t) \odot \boldsymbol{\delta}(t). \tag{6}$$

Using this notation we can rewrite the iteration scheme (4) as
$$\boldsymbol{v}(t+1) = \boldsymbol{v}(t) + \boldsymbol{d}(t) \quad (t = 0, 1, \cdots).$$
By reformulating (6) as
$$\boldsymbol{d}(t) = \boldsymbol{\delta}(t) + (\boldsymbol{\beta}(t) - \mathbf{1}) \odot \boldsymbol{\delta}(t),$$
where $\boldsymbol{\beta}(t) - \mathbf{1} \to \mathbf{0}$ as $t \to \infty$, we regard $\boldsymbol{\delta}(t)$ as the decisive part of $\boldsymbol{d}(t)$ and acquire properties of $\boldsymbol{d}(t)$ through exploring analogues of $\boldsymbol{\delta}(t)$.

First, we can show a basic estimation:
$$\boldsymbol{\delta}(t)^T \widehat{\boldsymbol{u}} = \|P\boldsymbol{\delta}(t)\| \geq \frac{\|\boldsymbol{\delta}(t)\|}{\max\limits_n \|\boldsymbol{\xi}_n\|} \quad (t = 0, 1, 2 \cdots).$$

The projection properties of $\boldsymbol{\delta}(t)$ is easily passed on to $\boldsymbol{d}(t)$. In fact, for sufficiently large $t$,
$$\boldsymbol{d}(t)^T \widehat{\boldsymbol{u}} = \|P\boldsymbol{d}(t)\| \geq \frac{\|\boldsymbol{d}(t)\|}{4 \max\limits_n \|\boldsymbol{\xi}_n\|}, \tag{7}$$

Inequality (7) provides a cumulative effect on the projection of $\boldsymbol{v}(t)$ as $t$ increases:
$$\|P\boldsymbol{v}(t)\| \geq \frac{\|\boldsymbol{v}(t)\|}{8\max\limits_n \|\boldsymbol{\xi}_n\|}, \quad \text{for sufficiently large } t.$$

The following lemma reveals a crucial characteristic of the iterative process (4): as $t$ tends to infinity, the contribution of $\boldsymbol{\delta}(t)$ to the increment of the deviation from the direction of $\widehat{\boldsymbol{u}}$, compared to its contribution to the increment in the direction of $\widehat{\boldsymbol{u}}$, becomes more and more insignificant.

**Lemma 3.5.** Given $\varepsilon > 0$. Let $a, b, c$ be positive numbers as defined in Assumption 3 in Section 2. If $\|Q\boldsymbol{v}(t)\| > 2N(c+1)(ace\varepsilon)^{-1}$, then for sufficiently large $t$,
$$Q\boldsymbol{v}(t)^T \boldsymbol{\delta}(t) < \varepsilon\|Q\boldsymbol{v}(t)\|\|\boldsymbol{\delta}(t)\|.$$

This property can be translated into a more convenient version for $\boldsymbol{d}(t)$.

**Lemma 3.6.** For any $\varepsilon > 0$, there exist $R > 0$ such that for sufficiently large $t$ and $\|Q\boldsymbol{v}(t)\| \geq R$,
$$\|Q\boldsymbol{v}(t+1)\| - \|Q\boldsymbol{v}(t)\| \leq \varepsilon\|\boldsymbol{d}(t)\|.$$

Therefore, over a long period, the cumulative increment of $\boldsymbol{v}(t)$ in the direction of $\widehat{\boldsymbol{u}}$ will overwhelm the deviation from it, yielding the existence of an asymptotic direction for $\boldsymbol{v}(t)$.

**Lemma 3.7.**
$$\lim_{t \to \infty} \frac{\boldsymbol{v}(t)}{\|\boldsymbol{v}(t)\|} = \widehat{\boldsymbol{u}}. \tag{8}$$

By the relation (3) between $\boldsymbol{v}(t)$ and $\boldsymbol{w}(t)$, our main result directly follows from (8).

**Theorem 3.2.** AdaGrad iterates (1) has an asymptotic direction:
$$\lim_{t \to \infty} \frac{\boldsymbol{w}(t)}{\|\boldsymbol{w}(t)\|} = \frac{\widetilde{\boldsymbol{w}}}{\|\widetilde{\boldsymbol{w}}\|},$$
where
$$\widetilde{\boldsymbol{w}} = \underset{\boldsymbol{w}^T \boldsymbol{x}_n \geq 1, \forall n}{\arg\min} \left\| \frac{1}{\sqrt{\boldsymbol{h}_\infty}} \odot \boldsymbol{w} \right\|^2. \tag{9}$$

### 3.3 Factors Affecting the Asymptotic Direction

Theorem 3.2 confirms that AdaGrad iterates (1) have an asymptotic direction $\widetilde{\boldsymbol{w}}/\|\widetilde{\boldsymbol{w}}\|$, where $\widetilde{\boldsymbol{w}}$ is the solution to the optimization problem (9). Since the objective function $\left\|\boldsymbol{h}_\infty^{-1/2} \odot \boldsymbol{w}\right\|^2$ is determined by the limit vector $\boldsymbol{h}_\infty$, it is easy to see that the asymptotic direction may depend on the choices of the dataset $\{(\boldsymbol{x}_n, y_n)\}_{n=1}^N$, the hyperparameters $\epsilon$, $\eta$, and the initial point $\boldsymbol{w}(0)$. In the following we will discuss this varied dependency in several respects.

### 3.3.1 Difference from the Asymptotic Direction of GD iterates

When the classic gradient descent method is applied to minimize the same loss, it is known (see Theorem 3, Soudry et al. [2018]) that GD iterates

$$\boldsymbol{w}_G(t+1) = \boldsymbol{w}_G(t) - \eta \nabla \mathcal{L}\left(\boldsymbol{w}_G(t)\right) \quad (t = 0, 1, 2, \cdots), \tag{10}$$

have an asymptotic direction $\widehat{\boldsymbol{w}}/\|\widehat{\boldsymbol{w}}\|$, where $\widehat{\boldsymbol{w}}$ is the solution of the hard max-margin SVM problem

$$\underset{\boldsymbol{w}^T \boldsymbol{x}_n \geq 1, \forall n}{\arg \min} \|\boldsymbol{w}\|^2. \tag{11}$$

The two optimization problems (9) and (11) have the same feasible set

$$\left\{\boldsymbol{w} \in \mathbb{R}^p : \boldsymbol{w}^T \boldsymbol{x}_n \geq 1, \text{ for } n = 1, \cdots, N \right\},$$

but they take on different objective functions. It is natural to expect that their solutions $\widetilde{\boldsymbol{w}}$ and $\widehat{\boldsymbol{w}}$ yield different directions, as shown in the following toy example.

**Example 3.1.** Let $\boldsymbol{x}_1 = (\cos\theta, \sin\theta)^T$ and $\mathcal{L}(\boldsymbol{w}) = e^{-\boldsymbol{w}^T \boldsymbol{x}_1}$. Suppose $0 < \theta < \pi/2$. In this setting we simply have $\widehat{\boldsymbol{w}} = \boldsymbol{x}_1$. Selecting $\boldsymbol{w}(0) = (a, b)^T$ and $\epsilon = 0$, we have

$$-\boldsymbol{g}(0) = e^{-\boldsymbol{w}(0)^T \boldsymbol{x}_1} \boldsymbol{x}_1 = e^{-a\cos\theta - b\sin\theta} (\cos\theta, \sin\theta)^T,$$

$$\boldsymbol{h}(0) = (h_1(0), h_2(0))^T = e^{a\cos\theta + b\sin\theta} \left(\frac{1}{\cos\theta}, \frac{1}{\sin\theta}\right)^T.$$

In general we can show there is a sequence of positive numbers $p(t)$ such that

$$-\boldsymbol{g}(t) = p(t) (\cos\theta, \sin\theta)^T,$$

and

$$\boldsymbol{h}_\infty = \lim_{t \to \infty} \frac{1}{\sqrt{p(0)^2 + p(1)^2 + \cdots + p(t)^2}} \left(\frac{1}{\cos\theta}, \frac{1}{\sin\theta}\right)^T = \frac{1}{\rho} \left(\frac{1}{\cos\theta}, \frac{1}{\sin\theta}\right)^T.$$

Now

$$\begin{aligned}
\widetilde{\boldsymbol{w}} &= \underset{\boldsymbol{w}^T \boldsymbol{x}_1 \geq 1}{\arg \min} \left\|\boldsymbol{h}_\infty^{-1/2} \odot \boldsymbol{w}\right\|^2 = \underset{\boldsymbol{w}^T \boldsymbol{x}_1 \geq 1}{\arg \min} \rho \left(w_1^2 \cos\theta + w_2^2 \sin\theta\right) \\
&= \underset{\boldsymbol{w}^T \boldsymbol{x}_1 \geq 1}{\arg \min} \left(w_1^2 \cos\theta + w_2^2 \sin\theta\right) = \left(\frac{1}{\cos\theta + \sin\theta}, \frac{1}{\cos\theta + \sin\theta}\right),
\end{aligned}$$

and we have $\widetilde{\boldsymbol{w}}/\|\widetilde{\boldsymbol{w}}\| = \left(\sqrt{2}/2, \sqrt{2}/2\right)^T$. Note that this direction is invariant when $\theta$ ranges between 0 and $\pi/2$, i.e., irrelevant to $\boldsymbol{x}_1$. These two directions coincide only when $\theta = \pi/4$.

### 3.3.2 Sensitivity to Small Coordinate System Rotations

If we consider the same setting as in Example 3.1, but taking $\theta \in (\pi/2, \pi)$. Then the asymptotic direction $\widetilde{\boldsymbol{w}}/\|\widetilde{\boldsymbol{w}}\|$ will become $\left(-\sqrt{2}/2, \sqrt{2}/2\right)^T$. This implies, however, if $\boldsymbol{x}_1$ is close to the direction of $y-$axis, then a small rotation of the coordinate system may result in a large change of the asymptotic direction reaching a right angle, i.e., in this case the asymptotic direction is highly unstable even for a small perturbation of its $x-$coordinate.

### 3.3.3 Effects of the Initialization and Hyperparameter $\eta$

It is reasonable to believe that the asymptotic direction of AdaGrad depends on the initial conditions, including initialization and step size (see Section 3.3, Gunasekar et al. [2018a]). Theorem 3.2 yields a geometric interpretation for this dependency as shown in Figure 1, where the red arrows indicate $\boldsymbol{x}_1 = (\cos(3\pi/8), \sin(3\pi/8))$ and $\boldsymbol{x}_2 = (\cos(9\pi/20), \sin(9\pi/20))$, and the cyan arrow indicates the max-margin separator, which points at $\boldsymbol{m}$, the corner of the feasible set $\left\{\boldsymbol{w} \mid \boldsymbol{w}^T \boldsymbol{x}_n \geq 1, \forall n = 1, 2\right\}$ (the yellow shadowed area).

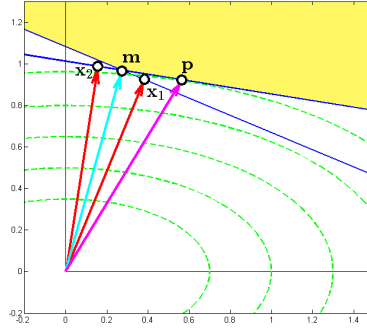

Figure 1: A case that the asymptotic directions of AdaGrad and GD are different.

Since the isolines of the function $\left\| \boldsymbol{h}_{\infty}^{-1/2} \odot \boldsymbol{w} \right\|^2$ are ellipses (the green dashed curves) centered at the origin, the unique minimizer of the function in the feasible set must be the tangency point $\boldsymbol{p}$ (pointed at by the magenta arrow) between the tangent ellipse and the boundary of the feasible set. If $\boldsymbol{h}_{\infty}$ varies, then the eccentricity of the tangent ellipses may change. It makes the tangency point move along the boundary, indicating the change of the asymptotic direction.

Numerical simulations also reveal the differences among the asymptotic directions of Ada-Grad iterates with various learning rates, as shown in Figure 2. On the left-hand diagram, $\mathbf{x}_1 = \left( \cos\left(\pi/8\right), \sin\left(\pi/8\right) \right)$ and $\mathbf{x}_2 = \left( \cos\left(\pi/20\right), \sin\left(\pi/20\right) \right)$ are two support vectors. $\boldsymbol{d}_m$ denotes the direction of the max-margin separator. $\boldsymbol{d}_{01}$ and $\boldsymbol{d}_{05}$ denote the directions of AdaGrad iterates computed after $10^8$ steps, with $\eta = 0.1$ and $0.5$, respectively. The small angle between the two may indicates that the asymptotic direction depend on $\eta$. However, all the asymptotic directions apparently diverge from the max-margin separator. On the right-hand diagram, the red and blue curves plot $\left\| \mathbf{w}(t)/\|\mathbf{w}(t)\| - \mathbf{d}_m \right\|$ vs. the number of the iterates with $\eta = 0.1$ and $0.5$, respectively. It illustrates that the two sequences of the directions of AdaGrad iterates slowly converge to their own asymptotic directions, slightly different from each other.

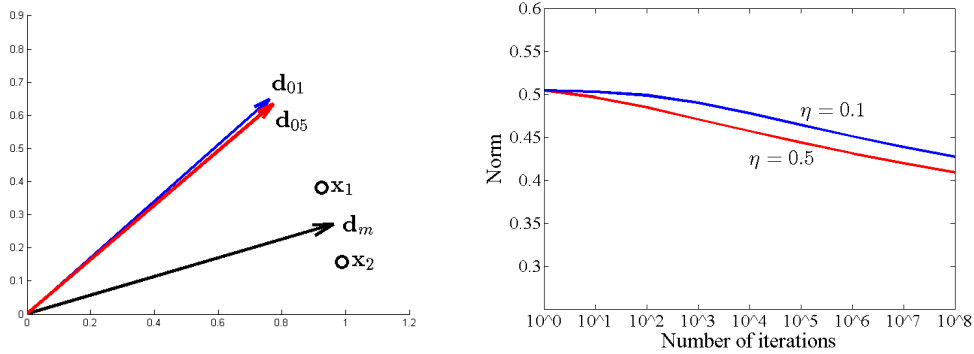

Figure 2: Numerical simulations with $\eta = 0.1$ and $0.5$.

### 3.3.4 Cases that the Asymptotic Direction is Stable

Above we have observed that the asymptotic direction of AdaGrad iterates can be very different from the asymptotic direction of GD iterates, which is robust with respect to different choices of initialization and learning rate $\eta$. It is natural to ask what are the conditions under which the two asymptotic directions coincide. The following proposition provides a sufficient one.

**Proposition 3.1.** Let $\boldsymbol{a} = (a_1, \cdots, a_p)^T$ be a vector satisfying $\boldsymbol{a}^T \boldsymbol{x}_n \geq 1 \ (n = 1, \cdots, N)$ and $a_1 \cdots a_p \neq 0$. Suppose that $\boldsymbol{w} = (w_1, \cdots, w_p)^T$ satisfies $\boldsymbol{w}^T \boldsymbol{x}_n \geq 1 \ (n = 1, \cdots, N)$ and

$$a_i \left( w_i - a_i \right) \geq 0 \ (i = 1, \cdots, p).$$

Then for any $\boldsymbol{b} = (b_1, \cdots, b_p)^T$ such that $b_1 \cdots b_p \neq 0$,

$$\underset{\boldsymbol{w}^T \boldsymbol{x}_n \geq 1, \forall n}{\arg\min} \ \|\boldsymbol{b} \odot \boldsymbol{w}\|^2 = \underset{\boldsymbol{w}^T \boldsymbol{x}_n \geq 1, \forall n}{\arg\min} \ \|\boldsymbol{w}\|^2 = \boldsymbol{a},$$

and therefore the asymptotic directions of AdaGrad (1) and GD (10) are equal.

Such a condition seems at fist sight quite harsh to be satified. However, there is a significant proportion of the chances that a dataset $\{\boldsymbol{x}_n : \ n = 1, \cdots, N\}$ meets the requirement, as shown in the following result.

**Proposition 3.2.** Suppose $N \geq p$ and $\boldsymbol{X} = [\boldsymbol{x}_1, \cdots, \boldsymbol{x}_N] \in \mathbb{R}^{p \times N}$ is sampled from any distribution whose density function is nonzero almost everywhere. Then with a positive probability the asymptotic directions of AdaGrad (1) and GD (10) are equal.

**Example 3.2.** Let $r_1, r_2 > 0$,

$$\boldsymbol{x}_1 = r_1 \left( \cos \theta_1, \sin \theta_1 \right)^T, \quad \frac{\pi}{2} \leq \theta_1 < \pi,$$

$$\boldsymbol{x}_2 = r_2 \left( \cos \theta_2, \sin \theta_2 \right)^T, \quad \theta_1 - \pi < \theta_2 \leq 0,$$

and $\mathcal{L}(\boldsymbol{w}) = l(\boldsymbol{w}^T \boldsymbol{x}_1) + l(\boldsymbol{w}^T \boldsymbol{x}_2)$. The system of equations

$$\boldsymbol{w}^T \boldsymbol{x}_i = 1 \ (i = 1, 2)$$

has a unique solution $(\alpha, \beta)^T$, where

$$\alpha = \frac{r_2^{-1} \sin \theta_1 - r_1^{-1} \sin \theta_2}{\sin (\theta_1 - \theta_2)} > 0, \quad \beta = \frac{r_1^{-1} \cos \theta_2 - r_2^{-1} \cos \theta_1}{\sin (\theta_1 - \theta_2)} > 0.$$

It is easy to check that if $\boldsymbol{w} = (w_1, w_2)^T$ satisfies $\boldsymbol{w}^T \boldsymbol{x}_i \geq 1 \ (i = 1, 2)$, then $w_1 \geq \alpha$, $w_2 \geq \beta$. Thus any quadratic form $b_1 w_1^2 + b_2 w_2^2 \ (b_1, b_2 > 0)$ takes its minimum at $(\alpha, \beta)^T$ over the feasible set $\{\boldsymbol{w} : \ \boldsymbol{w}^T \boldsymbol{x}_i \geq 1 \ (i = 1, 2)\}$. Hence the asymptotic direction of AdaGrad (1) applying to this problem is always equal to $(\alpha, \beta)^T / \|(\alpha, \beta)\|$, which is also the asymptotic direction of GD (10).

A geometric perspective of this example is given in Figure 2, where the red arrows indicate $\boldsymbol{x}_1 = \left( \cos (5\pi/8), \sin (5\pi/8) \right)$ and $\boldsymbol{x}_2 = \left( \cos (-\pi/8), \sin (-\pi/8) \right)$, and the magenta arrow indicates $(\alpha, \beta)^T$. It is easy to see that the isoline (the thick ellipse drawn in green) along which the function $\left\| \boldsymbol{h}_\infty^{-1/2} \odot \boldsymbol{w} \right\|^2$ equals its minimum must intersect with the feasible set (the grey shadowed area) at the corner $(\alpha, \beta)^T$, no matter what $\boldsymbol{h}_\infty$ is.

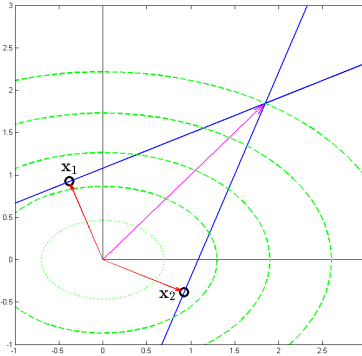

Figure 3: A case that the asymptotic directions of AdaGrad and GD are equal.

# 4 Conclusion

We proved that the basic diagonal AdaGrad, when minimizing a smooth monotone loss function with an exponential tail, has an asymptotic direction, which can be characterized as the solution of a quadratic optimization problem. In this respect AdaGrad is similar to GD, even though their asymptotic directions are usually different. The difference between them also lies in the stability of their asymptotic directions. The asymptotic direction of GD is uniquely determined by the predictors $x_n$'s and independent of initialization and the learning rate, as well as the rotation of coordinate system, while the asymptotic direction of AdaGrad is likely to be affected by those factors.

In spite of all these findings, we still do not know whether the asymptotic direction of AdaGrad will change for various initialization or different learning rates. Furthermore, we hope our approach can be applied to the research on the implicit biases of other adaptive methods such as AdaDelta, RMSProp, and Adam.

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
