[Supplementary Material]

# The Implicit Bias of AdaGrad on Separable Data

**Qian Qian**
Department of Statistics
The Ohio State University
Columbus, OH 43210, USA
qian.216@osu.edu

**Xiaoyuan Qian**
School of Mathematical Sciences
Dalian University of Technology
Dalian, Liaoning 116024, China
xyqian@dlut.edu.cn

## Appendix

To simplify notation, we denote

$$S_i(t) = \sum_{\tau=0}^{t} g_i(\tau)^2 \,,$$

for all $i \in \{1, \cdots, p\}$ and $t = 0, 1, 2, \cdots$.

**Lemma 3.1.** $\mathcal{L}\left(\boldsymbol{w}(t+1)\right) < \mathcal{L}\left(\boldsymbol{w}(t)\right) \;\; (\, t = 0, 1, \cdots)$.

*Proof.* Since $l$ is $\beta-$smooth, so is $\mathcal{L}$. Thus we have

$$
\begin{aligned}
&\mathcal{L}(\boldsymbol{w}(t+1)) \\
\leq \;& \mathcal{L}(\boldsymbol{w}(t)) + \nabla\mathcal{L}(\boldsymbol{w}(t))\left(\boldsymbol{w}(t+1) - \boldsymbol{w}(t)\right) \\
&+ \frac{\beta}{2}\|\boldsymbol{w}(t+1) - \boldsymbol{w}(t)\|^2 \\
= \;& \mathcal{L}(\boldsymbol{w}(t)) - \eta \boldsymbol{g}(t)^T\left(\boldsymbol{h}(t) \odot \boldsymbol{g}(t)\right) \\
&+ \frac{\beta\eta^2}{2}\|\boldsymbol{h}(t) \odot \boldsymbol{g}(t)\|^2 \,.
\end{aligned}
$$

Thus

$$
\begin{aligned}
&\mathcal{L}(\boldsymbol{w}(t)) - \mathcal{L}(\boldsymbol{w}(t+1)) \\
\geq \;& \eta \boldsymbol{g}(t)^T\left(\boldsymbol{h}(t) \odot \boldsymbol{g}(t)\right) - \frac{\beta\eta^2}{2}\|\boldsymbol{h}(t) \odot \boldsymbol{g}(t)\|^2 \\
= \;& \eta \sum_{i=1}^{p} \frac{g_i(t)^2}{\sqrt{S_i(t) + \epsilon}} - \frac{\beta\eta^2}{2}\sum_{i=1}^{p}\frac{g_i(t)^2}{S_i(t) + \epsilon} \\
= \;& \eta \sum_{i=1}^{p}\left(1 - \frac{\beta\eta}{2\sqrt{S_i(t) + \epsilon}}\right)\frac{g_i(t)^2}{\sqrt{S_i(t) + \epsilon}} \\
> \;& 0. 
\end{aligned}
\tag{1}
$$

**Lemma 3.2.** $\sum_{t=0}^{\infty}\|\boldsymbol{g}(t)\|^2 < \infty$.

*Proof.* We use reduction of absurdity. Suppose

$$\sum_{t=1}^{\infty}\|\boldsymbol{g}(t)\|^2 \;=\; \infty\,.$$

Then there is some $k \in \{1, \cdots, p\}$ such that

$$\lim_{t \to \infty} S_k(t) = \sum_{t=1}^{\infty} g_k(t)^2 = \infty \,. \tag{2}$$

Thus we can find a time $t_0$ such that, for all $t > t_0$,

$$S_i(t) > \max(\beta\eta, \, 1) \,.$$

Noting that positive series

$$\sum_{t=1}^{\infty} a_t \,, \quad \sum_{t=1}^{\infty} \frac{a_t}{a_1 + \cdots + a_t + \epsilon}$$

converge or diverge simultaneously, so we obtain from (2)

$$\sum_{t=0}^{\infty} \frac{g_k(t)^2}{S_k(t) + \epsilon} = \infty \,.$$

Therefore,

$$\sum_{t=0}^{\infty} \left( 1 - \frac{\beta\eta}{2\sqrt{S_k(t) + \epsilon}} \right) \frac{g_k(\tau)^2}{\sqrt{S_k(t) + \epsilon}}$$

$$= \sum_{t=0}^{t_0} \left( 1 - \frac{\beta\eta}{2\sqrt{S_k(t) + \epsilon}} \right) \frac{g_k(\tau)^2}{\sqrt{S_k(t) + \epsilon}}$$

$$+ \sum_{t>t_0} \left( 1 - \frac{\beta\eta}{2\sqrt{S_k(t) + \epsilon}} \right) \frac{g_k(\tau)^2}{\sqrt{S_k(t) + \epsilon}}$$

$$\geq \quad C + \frac{1}{2} \sum_{t>t_0} \frac{g_k(t)^2}{\sqrt{S_k(t) + \epsilon}}$$

$$\geq \quad C + \frac{1}{2} \sum_{t>t_0} \frac{g_k(t)^2}{S_k(t) + \epsilon}$$

$$= \quad C + \infty = \infty \,, \tag{3}$$

where the constant

$$C = \sum_{t=0}^{t_0} \left( 1 - \frac{\beta\eta}{2\sqrt{S_k(t) + \epsilon}} \right) \frac{g_k(\tau)^2}{\sqrt{S_k(t) + \epsilon}} \,.$$

On the other hand, from (1) we have

$$\sum_{\tau=0}^{t} \left( 1 - \frac{\beta\eta}{2\sqrt{S_k(t) + \epsilon}} \right) \frac{g_k(\tau)^2}{\sqrt{S_k(t) + \epsilon}}$$

$$\leq \quad \sum_{\tau=0}^{t} \sum_{i=1}^{p} \left( 1 - \frac{\beta\eta}{2\sqrt{S_i(\tau) + \epsilon}} \right) \frac{g_i(\tau)^2}{\sqrt{S_i(\tau) + \epsilon}}$$

$$= \quad \sum_{i=1}^{p} \left\{ \sum_{\tau=0}^{t} \left( 1 - \frac{\beta\eta}{2\sqrt{S_i(\tau) + \epsilon}} \right) \frac{g_i(\tau)^2}{\sqrt{S_i(\tau) + \epsilon}} \right\}$$

$$\leq \quad \frac{1}{\eta} \left( \mathcal{L}(\boldsymbol{w}(0)) - \mathcal{L}(\boldsymbol{w}(t+1)) \right) \leq \frac{\mathcal{L}(\boldsymbol{w}(0))}{\eta} \,,$$

implying, for sufficiently small $\eta$,

$$\sum_{t=0}^{\infty} \left( 1 - \frac{\beta\eta}{2\sqrt{S_k(t) + \epsilon}} \right) \frac{g_k(\tau)^2}{\sqrt{S_k(t) + \epsilon}} \leq \frac{\mathcal{L}(\boldsymbol{w}(0))}{\eta},$$

which contradicts to (3).

**Lemma 3.3.** The following statements hold:

(i) $\|\boldsymbol{g}(t)\| \to 0 \; (t \to \infty)$.

(ii) $\|\boldsymbol{w}(t)\| \to \infty \; (t \to \infty)$.

(iii) $\mathcal{L}(\boldsymbol{w}(t)) \to 0 \; (t \to \infty)$.

(iv) $\forall n, \; \lim_{t \to \infty} \boldsymbol{w}(t)^T \boldsymbol{x}_n = \infty$.

(v) $\exists t_0, \; \forall t > t_0, \; \boldsymbol{w}(t)^T \boldsymbol{x}_n > 0$.

*Proof.* Lemma 3.2 implies (i), which yields (ii).

To prove (iii), we use reduction to absurdity. Assume

$$\overline{\lim_{t \to \infty}} \mathcal{L}(\boldsymbol{w}(t)) = c > 0.$$

Then there exists an index $m \in \{1, \cdots, N\}$ such that

$$\overline{\lim_{t \to \infty}} \, l\left(\boldsymbol{w}(t)^T \boldsymbol{x}_m\right) \geq \frac{c}{N} > 0.$$

By Assumption 2, we have $l(u) \to 0 \; (u \to \infty)$. Thus we can find a constant $M > 0$ such that

$$\underline{\lim_{t \to \infty}} \, \boldsymbol{w}(t)^T \boldsymbol{x}_m \leq M,$$

which implies that there exists a sequence of times

$$t_1 < t_2 < t_3 < \cdots$$

such that

$$\lim_{k \to \infty} \boldsymbol{w}\left(t_k\right)^T \boldsymbol{x}_m = \gamma \leq M.$$

Choose a vector $\boldsymbol{w}_*$ such that $\boldsymbol{w}_*^T \boldsymbol{x}_n > 0$ for all $n \in \{1, \cdots, N\}$. Noting that $-l' > 0$, we have

$$
\begin{aligned}
-\boldsymbol{w}_*^T \boldsymbol{g}(t) &= -\sum_{n=1}^{N} l'\left(\boldsymbol{w}(t)^T \boldsymbol{x}_n\right) \boldsymbol{w}_*^T \boldsymbol{x}_n \\
&\geq l'\left(\boldsymbol{w}(t)^T \boldsymbol{x}_m\right) \boldsymbol{w}_*^T \boldsymbol{x}_m
\end{aligned}
$$

Thus

$$
\begin{aligned}
\overline{\lim_{k \to \infty}} &- \boldsymbol{w}_*^T \boldsymbol{g}\left(t_k\right) \\
&\geq \overline{\lim_{k \to \infty}} \, l'\left(\boldsymbol{w}\left(t_k\right)^T \boldsymbol{x}_m\right) \boldsymbol{w}_*^T \boldsymbol{x}_m \\
&= \left(\boldsymbol{w}_*^T \boldsymbol{x}_m\right) \lim_{k \to \infty} l'\left(\boldsymbol{w}\left(t_k\right)^T \boldsymbol{x}_m\right) \\
&= \left(\boldsymbol{w}_*^T \boldsymbol{x}_m\right) l'\left(\gamma\right) > 0.
\end{aligned}
\tag{4}
$$

Note that

$$\|\boldsymbol{g}\left(t\right)\| \to 0 \; (t \to \infty)$$

implies

$$-\boldsymbol{w}_*^T \boldsymbol{g}\left(t_k\right) \leq \|\boldsymbol{w}_*\| \, \|\boldsymbol{g}\left(t_k\right)\| \to 0 \; (k \to \infty),$$

which contradicts to (4), meaning (iii) has to be true.

(iv) follows from (iii). (v) follows directly from (iv).

**Theorem 3.1.** The sequence $\{\boldsymbol{h}(t)\}_{t=0}^{\infty}$ converges as $t \to \infty$ to a vector

$$\boldsymbol{h}_\infty = (h_{\infty,1}, \cdots, h_{\infty,p})$$

satisfying $h_{\infty,i} > 0 \; (i = 1, \cdots, p)$.

*Proof.* By Lemma 3.2 $\{h_i(t)\}_{t=0}^{\infty}$ is decreasing and has a lower bound

$$\frac{1}{\sqrt{S + \epsilon}} > 0,$$

where
$$S = \lim_{t \to \infty} S_i(t) \le \sum_{t=0}^{\infty} \|\boldsymbol{g}(t)\|^2 < \infty,$$
then converges, for each $i \in \{1, \cdots, p\}$.

**Lemma A.1.** Let $\boldsymbol{a}$, $\boldsymbol{b} = (b_1, \cdots, b_p) \in \mathbb{R}^p$. Then the following relations hold.

(i) Associativity. $(\boldsymbol{a} \odot \boldsymbol{b}) \odot \boldsymbol{v} = \boldsymbol{a} \odot (\boldsymbol{b} \odot \boldsymbol{v})$.

(ii) Commutativity. $\boldsymbol{a} \odot \boldsymbol{b} = \boldsymbol{b} \odot \boldsymbol{a}$;

(iii) Distributivity. $\boldsymbol{a} \odot (\boldsymbol{b} + \boldsymbol{c}) = \boldsymbol{a} \odot \boldsymbol{b} + \boldsymbol{a} \odot \boldsymbol{c}$.

(iv) $\min_i |b_i| \|\boldsymbol{a}\| \le \|\boldsymbol{b} \odot \boldsymbol{a}\| \le \max_i |b_i| \|\boldsymbol{a}\|$.

*Proof.* Obviously.

**Lemma 3.4.** The following statements hold:

(i) $\|\nabla \mathcal{L}_{ind}(t)\| \to 0 \ (t \to \infty)$.

(ii) $\|\boldsymbol{v}(t)\| \to \infty \ (t \to \infty)$.

(iii) $\mathcal{L}_{ind}(\boldsymbol{v}(t)) \to 0 \ (t \to \infty)$.

(iv) $\forall n, \ \lim_{t \to \infty} \boldsymbol{v}(t)^T \boldsymbol{\xi}_n = \infty$.

(v) $\exists t_0, \ \forall t > t_0, \ \boldsymbol{v}(t)^T \boldsymbol{\xi}_n > 0$.

*Proof.* It directly follows from Lemma 3.3.

**Lemma A.2.** For $t = 0, 1, 2 \cdots$,
$$\boldsymbol{\delta}(t)^T \widehat{\boldsymbol{u}} = \|P\boldsymbol{\delta}(t)\| \ge \frac{\|\boldsymbol{\delta}(t)\|}{\max_n \|\boldsymbol{\xi}_n\|},$$
where
$$\widehat{\boldsymbol{u}} = \underset{\boldsymbol{u}^T \boldsymbol{\xi}_n \ge 1, \forall n}{\arg \min} \|\boldsymbol{u}\|^2.$$

*Proof.* From Assumption 2. we have $-l'\left(\boldsymbol{v}^T \boldsymbol{\xi}_n\right) > 0$. By the definition of $\widehat{\boldsymbol{u}}$ we have
$$\boldsymbol{\xi}_n^T \widehat{\boldsymbol{u}} \ge 1 \quad (n = 1, \cdots, N).$$
Thus
$$\boldsymbol{\delta}(t)^T \widehat{\boldsymbol{u}} = -\eta \nabla \mathcal{L}_{ind}\left(\boldsymbol{v}(t)\right)^T \widehat{\boldsymbol{u}} = -\eta \sum_{n=1}^{N} l'\left(\boldsymbol{v}^T \boldsymbol{\xi}_n\right) \boldsymbol{\xi}_n^T \widehat{\boldsymbol{u}} \ge -\eta \sum_{n=1}^{N} l'\left(\boldsymbol{v}^T \boldsymbol{\xi}_n\right) > 0.$$

Note that $l' < 0$. We have
$$
\begin{aligned}
\|\boldsymbol{\delta}(t)\| &= \left\| \eta \sum_{n=1}^{N} l'\left(\boldsymbol{v}(t)^T \boldsymbol{\xi}_n\right) \boldsymbol{\xi}_n \right\| \le -\eta \sum_{n=1}^{N} l'\left(\boldsymbol{v}(t)^T \boldsymbol{\xi}_n\right) \|\boldsymbol{\xi}_n\| \\
&\le \max_n \|\boldsymbol{\xi}_n\| \left( -\eta \sum_{n=1}^{N} l'\left(\boldsymbol{v}(t)^T \boldsymbol{\xi}_n\right) \right),
\end{aligned}
$$
or
$$-\eta \sum_{n=1}^{N} l'\left(\boldsymbol{v}(t)^T \boldsymbol{\xi}_n\right) \ge \frac{\|\boldsymbol{\delta}(t)\|}{\max_n \|\boldsymbol{\xi}_n\|}. \tag{5}$$

On the other hand,
$$
\begin{aligned}
P\boldsymbol{\delta}(t) &= -\eta P \sum_{n=1}^{N} l'\left(\boldsymbol{v}(t)^T \boldsymbol{\xi}_n\right) \boldsymbol{\xi}_n = -\eta \sum_{n=1}^{N} l'\left(\boldsymbol{v}(t)^T \boldsymbol{\xi}_n\right) P\boldsymbol{\xi}_n \\
&= -\eta \sum_{n=1}^{N} l'\left(\boldsymbol{v}(t)^T \boldsymbol{\xi}_n\right) \left(\boldsymbol{\xi}_n^T \widehat{\boldsymbol{u}}\right) \widehat{\boldsymbol{u}}.
\end{aligned}
$$

Noting $\boldsymbol{\xi}_n^T \widehat{\boldsymbol{u}} \geq 1$ $(n \in \{1, \cdots, N\})$, from (5) we obtain

$$\|P\boldsymbol{\delta}(t)\| = -\eta \sum_{n=1}^{N} l' \left(\boldsymbol{v}(t)^T \boldsymbol{\xi}_n\right) \left(\boldsymbol{\xi}_n^T \widehat{\boldsymbol{u}}\right) \geq -\eta \sum_{n=1}^{N} l' \left(\boldsymbol{v}(t)^T \boldsymbol{\xi}_n\right) \geq \frac{\|\boldsymbol{\delta}(t)\|}{\max\limits_{n} \|\boldsymbol{\xi}_n\|} .$$

**Lemma A.3.** For sufficiently large $t$,

$$\frac{1}{2} \|\boldsymbol{\delta}(t)\| \leq \|\boldsymbol{d}(t)\| \leq \frac{3}{2} \|\boldsymbol{\delta}(t)\|, \tag{6}$$

$$\|P\boldsymbol{d}(t)\| \geq \frac{\|\boldsymbol{d}(t)\|}{4 \max\limits_{n} \|\boldsymbol{\xi}_n\|} , \tag{7}$$

$$\boldsymbol{d}(t)^T \widehat{\boldsymbol{u}} = \|P\boldsymbol{d}(t)\| > 0 . \tag{8}$$

*Proof.* Let $\boldsymbol{\beta}(t) = (\beta_1(t), \cdots, \beta_p(t))^T$. Noting that

$$\|\boldsymbol{\beta}(t) - \mathbf{1}\| \to 0 \ \ (t \to \infty),$$

we can find some $t_0$ such that for $t \geq t_0$,

$$\frac{1}{2} \leq \min_{i} |\beta_i(t)| \leq \max_{i} |\beta_i(t)| \leq \frac{3}{2} \tag{9}$$

and

$$\max_{i} |\beta_i(t) - 1| < \frac{1}{2 \max\limits_{n} \|\boldsymbol{\xi}_n\|} . \tag{10}$$

The inequality (6) follows directly from (9). On the other hand,

$$P\boldsymbol{d}(t) = P \left(\boldsymbol{\beta}(t) \odot \boldsymbol{\delta}(t)\right) = P\boldsymbol{\delta}(t) + P \left((\boldsymbol{\beta}(t) - \mathbf{1}) \odot \boldsymbol{\delta}(t)\right)$$

By (10) we have

$$\|P \left((\boldsymbol{\beta}(t) - \mathbf{1}) \odot \boldsymbol{\delta}(t)\right)\| \leq \|(\boldsymbol{\beta}(t) - \mathbf{1}) \odot \boldsymbol{\delta}(t)\| \leq \max_{i} |\beta_i(t) - 1| \|\boldsymbol{\delta}(t)\| \leq \frac{\|\boldsymbol{\delta}(t)\|}{2 \max\limits_{n} \|\boldsymbol{\xi}_n\|} .$$

Hence

$$\begin{aligned}
\|P\boldsymbol{d}(t)\| &= \|P\boldsymbol{\delta}(t) + P \left((\boldsymbol{\beta}(t) - \mathbf{1}) \odot \boldsymbol{\delta}(t)\right)\| \\
&\geq \|P\boldsymbol{\delta}(t)\| - \|P \left((\boldsymbol{\beta}(t) - \mathbf{1}) \odot \boldsymbol{\delta}(t)\right)\| \\
&\geq \frac{\|\boldsymbol{\delta}(t)\|}{\max\limits_{n} \|\boldsymbol{\xi}_n\|} - \frac{\|\boldsymbol{\delta}(t)\|}{2 \max\limits_{n} \|\boldsymbol{\xi}_n\|} = \frac{\|\boldsymbol{\delta}(t)\|}{2\max\limits_{n} \|\boldsymbol{\xi}_n\|} .
\end{aligned}$$

Thus (7) follows from the left part of (6).

Noting that

$$\begin{aligned}
\boldsymbol{d}(t)^T \widehat{\boldsymbol{u}} &= \boldsymbol{\delta}(t)^T \widehat{\boldsymbol{u}} + (\boldsymbol{\beta}(t) - \mathbf{1}) \odot \boldsymbol{\delta}(t)^T \widehat{\boldsymbol{u}} \\
&\geq \boldsymbol{\delta}(t)^T \widehat{\boldsymbol{u}} - \left|(\boldsymbol{\beta}(t) - \mathbf{1}) \odot \boldsymbol{\delta}(t)^T \widehat{\boldsymbol{u}}\right| \\
&= \|P\boldsymbol{\delta}(t)\| - \|P \left((\boldsymbol{\beta}(t) - \mathbf{1}) \odot \boldsymbol{\delta}(t)\right)\| \\
&\geq \frac{\|\boldsymbol{\delta}(t)\|}{\max\limits_{n} \|\boldsymbol{\xi}_n\|} - \frac{\|\boldsymbol{\delta}(t)\|}{2\max\limits_{n} \|\boldsymbol{\xi}_n\|} \\
&= \frac{\|\boldsymbol{\delta}(t)\|}{2\max\limits_{n} \|\boldsymbol{\xi}_n\|} > 0 ,
\end{aligned}$$

we obtain (8).

**Lemma A.4.** For sufficiently large $t$,

$$\|P\boldsymbol{v}(t)\| \geq \frac{\|\boldsymbol{v}(t)\|}{8\max_n \|\boldsymbol{\xi}_n\|}.$$

*Proof.* By Lemma A.3 there there exists some $t_0$ such that for $t \geq t_0$,

$$\|P\boldsymbol{d}(t)\| \geq \frac{\|\boldsymbol{d}(t)\|}{4\max_n \|\boldsymbol{\xi}_n\|}.$$

Note that $\|\boldsymbol{v}(t)\| \to \infty$, which implies

$$\|\boldsymbol{d}(t_0)\| + \cdots + \|\boldsymbol{d}(t)\| \geq \|\boldsymbol{v}(t)\| - \|\boldsymbol{v}(t_0)\| \to \infty \ (t \to \infty).$$

Thus there exists some $t_1 > t_0$ such that for $t > t_1$,

$$\|\boldsymbol{d}(t_0)\| + \cdots + \|\boldsymbol{d}(t)\| > 2\|\boldsymbol{v}(t_0)\|,$$

Hence, meanwhile,

$$
\begin{aligned}
\|P\boldsymbol{v}(t)\| &= \|P\boldsymbol{v}(t_0) + P\boldsymbol{d}(t_0) + \cdots + P\boldsymbol{d}(t-1)\| \\
&= \|P\boldsymbol{v}(t_0)\| + \|P\boldsymbol{d}(t_0)\| + \cdots + \|P\boldsymbol{d}(t-1)\| \\
&\geq \frac{1}{4\max_n \|\boldsymbol{\xi}_n\|} \left(\|\boldsymbol{d}(t_0)\| + \cdots + \|\boldsymbol{d}(t-1)\|\right) \\
&\geq \frac{1}{8\max_n \|\boldsymbol{\xi}_n\|} \left(\|\boldsymbol{v}(t_0)\| + \|\boldsymbol{d}(t_0)\| + \cdots + \|\boldsymbol{d}(t-1)\|\right) \\
&\geq \frac{1}{8\max_n \|\boldsymbol{\xi}_n\|} \|\boldsymbol{v}(t_0) + \boldsymbol{d}(t_0) + \cdots + \boldsymbol{d}(t-1)\| \\
&= \frac{\|\boldsymbol{v}(t)\|}{8\max_n \|\boldsymbol{\xi}_n\|}.
\end{aligned}
$$

**Lemma A.5.** Let

$$\mathcal{K} = \left\{n : \ \boldsymbol{\xi}_n^T \widehat{\boldsymbol{u}} = 1\right\}.$$

Then there is a set of nonnegative numbers $\{\alpha_n : \ n \in \mathcal{K}\}$ such that

$$\widehat{\boldsymbol{u}} = \sum_{n \in \mathcal{K}} \alpha_n \boldsymbol{\xi}_n.$$

*Proof.* This is Lemma 12 in Appendix B of Soudry et al., [2018].

**Lemma 3.5.** Given $\varepsilon > 0$. Let $a, b, c$ be positive numbers as defined in Assumption 3 in Section 2. If $\|Q\boldsymbol{v}(t)\| > 2N(c+1)(ace\varepsilon)^{-1}$, then for sufficiently large $t$,

$$Q\boldsymbol{v}(t)^T \boldsymbol{\delta}(t) < \varepsilon \|Q\boldsymbol{v}(t)\| \|\boldsymbol{\delta}(t)\|.$$

*Proof.* Since for each $n \in \{1, \cdots, N\}$,

$$\boldsymbol{v}(t)^T \boldsymbol{\xi}_n \to \infty \ (t \to \infty),$$

we have, for sufficiently large $t$,

$$
\begin{aligned}
-l'\left(\boldsymbol{v}(t)^T \boldsymbol{\xi}_n\right) &= ce^{-a\boldsymbol{v}(t)^T \boldsymbol{\xi}_n} - r\left(\boldsymbol{v}(t)^T \boldsymbol{\xi}_n\right) \\
&\geq ce^{-a\boldsymbol{v}(t)^T \boldsymbol{\xi}_n} - e^{-(a+b)\boldsymbol{v}(t)^T \boldsymbol{\xi}_n} \\
&= e^{-a\boldsymbol{v}(t)^T \boldsymbol{\xi}_n}\left(c - e^{-b\boldsymbol{v}(t)^T \boldsymbol{\xi}_n}\right) \\
&= \frac{c}{2}e^{-a\boldsymbol{v}(t)^T \boldsymbol{\xi}_n}.
\end{aligned}
\tag{11}
$$

Similarly, we can prove for sufficiently large $t$,

$$-l'\left(\boldsymbol{v}(t)^T \boldsymbol{\xi}_n\right) \le 2ce^{-a\boldsymbol{v}(t)^T \boldsymbol{\xi}_n}. \tag{12}$$

Denote

$$p(t) = \boldsymbol{v}(t)^T \widehat{\boldsymbol{u}}, \quad \boldsymbol{q}(t) = Q\boldsymbol{v}(t).$$

Thus we have

$$\boldsymbol{v}(t) = p(t)\widehat{\boldsymbol{u}} + \boldsymbol{q}(t).$$

Denote

$$u_n = \widehat{\boldsymbol{u}}^T \boldsymbol{\xi}_n, \quad q_{n,t} = \boldsymbol{q}(t)^T \boldsymbol{\xi}_n.$$

We then have

$$
\begin{aligned}
& \boldsymbol{q}(t)^T \boldsymbol{\delta}(t) \\
= \ & -\eta \boldsymbol{q}(t)^T \sum_{n=1}^{N} l'\left(\boldsymbol{v}(t)^T \boldsymbol{\xi}_n\right) \boldsymbol{\xi}_n \\
= \ & -\eta \sum_{n=1}^{N} l'\left(\boldsymbol{v}(t)^T \boldsymbol{\xi}_n\right) \boldsymbol{q}(t)^T \boldsymbol{\xi}_n \\
= \ & -\eta \sum_{n=1}^{N} l'\left(\boldsymbol{v}(t)^T \boldsymbol{\xi}_n\right) q_{n,t} \\
\le \ & -\eta \sum_{n:\, q_{n,t}>0} l'\left(\boldsymbol{v}(t)^T \boldsymbol{\xi}_n\right) q_{n,t}.
\end{aligned}
$$

Applying (12) we obtain

$$
\begin{aligned}
\boldsymbol{q}(t)^T \boldsymbol{\delta}(t) \ \le \ & \eta \sum_{n:\, q_{n,t}>0} 2ce^{-a\boldsymbol{v}(t)^T \boldsymbol{\xi}_n} q_{n,t} \\
= \ & 2c\eta \sum_{n:\, q_{n,t}>0} e^{-a(p(t)\widehat{\boldsymbol{u}}+\boldsymbol{q}(t))^T \boldsymbol{\xi}_n} q_{n,t} \\
= \ & 2c\eta \sum_{n:\, q_{n,t}>0} e^{-ap(t)u_n} e^{-aq_{n,t}} q_{n,t} \\
\le \ & \frac{2c\eta}{ae} \sum_{n:\, q_{n,t}>0} e^{-ap(t)u_n} \\
\le \ & \frac{2c\eta N}{ae} e^{-ap(t)}.
\end{aligned}
$$

The last step is derived from $u_n \ge 1$ for $n \in \{1, \cdots, N\}$.

On the other hand, by Lemma A.5 there is a set of positive coefficients $\{\alpha_n : n \in \mathcal{K}\}$, where $\mathcal{K} = \{n:\ u_n = 1\}$, such that

$$\widehat{\boldsymbol{u}} = \sum_{n \in \mathcal{K}} \alpha_n \boldsymbol{\xi}_n .$$

Thus

$$0 = \boldsymbol{q}(t)^T \widehat{\boldsymbol{u}} = \sum_{n \in \mathcal{K}} \alpha_n \boldsymbol{q}(t)^T \boldsymbol{\xi}_n ,$$

implying there is at least one index $k \in \mathcal{K}$ such that

$$q_{t,k} = \boldsymbol{q}(t)^T \boldsymbol{\xi}_k \le 0 .$$

Hence

$$
\begin{aligned}
\|P\boldsymbol{\delta}(t)\| &= \eta \left\| P \sum_{n=1}^{N} l' \left( \boldsymbol{v}(t)^T \boldsymbol{\xi}_n \right) \boldsymbol{\xi}_n \right\| \\
&= \eta \left\| \sum_{n=1}^{N} l' \left( \boldsymbol{v}(t)^T \boldsymbol{\xi}_n \right) P\boldsymbol{\xi}_n \right\| \\
&= \eta \left\| \sum_{n=1}^{N} l' \left( \boldsymbol{v}(t)^T \boldsymbol{\xi}_n \right) u_n \widehat{\boldsymbol{u}} \right\| \\
&= -\eta \sum_{n=1}^{N} u_n l' \left( \boldsymbol{v}(t)^T \boldsymbol{\xi}_n \right) \\
&\geq -\eta \sum_{n=1}^{N} l' \left( \boldsymbol{v}(t)^T \boldsymbol{\xi}_n \right) \\
&> -\eta l' \left( \boldsymbol{v}(t)^T \boldsymbol{\xi}_k \right) .
\end{aligned}
$$

Noting that $u_k \geq 1$, $q_{t,k} \leq 0$, and the estimation (11), we obtain

$$
\begin{aligned}
\|P\boldsymbol{\delta}(t)\| &> -\eta l' \left( \boldsymbol{v}(t)^T \boldsymbol{\xi}_k \right) \\
&\geq \frac{c\eta}{2} e^{-a\boldsymbol{v}(t)^T \boldsymbol{\xi}_k} \\
&= \frac{c\eta}{2} e^{-ap(t)u_k} e^{-aq_{t,k}} \\
&\geq \frac{c\eta}{2} e^{-ap(t)}.
\end{aligned}
$$

Thus

$$
\begin{aligned}
\boldsymbol{q}(t)^T \boldsymbol{\delta}(t) &\leq \eta (c+1) \frac{N}{ae} e^{-ap(t)} \\
&\leq \frac{2N (c+1)}{ace} \|P\boldsymbol{\delta}(t)\| \\
&\leq \frac{2N (c+1)}{ace} \|\boldsymbol{\delta}(t)\| \\
&< \varepsilon \|\boldsymbol{q}(t)\| \|\boldsymbol{\delta}(t)\| ,
\end{aligned}
$$

for $\|\boldsymbol{q}(t)\| > 2(ace\varepsilon)^{-1} N(c+1)$.

**Lemma 3.6.** For any $\varepsilon > 0$, there exist $R > 0$ such that for sufficiently large $t$ and $\|Q\boldsymbol{v}(t)\| \geq R$,

$$
\|Q\boldsymbol{v}(t+1)\| - \|Q\boldsymbol{v}(t)\| \leq \varepsilon \|\boldsymbol{d}(t)\| .
$$

*Proof.* Again we denote $\boldsymbol{q}(t) = Q\boldsymbol{v}(t)$. By Lemma 3.5 we can choose a number $R > 0$ such that for sufficiently large $t$ and $\|\boldsymbol{q}(t)\| \geq R$,

$$
\boldsymbol{q}(t)^T \boldsymbol{\delta}(t) < \frac{\varepsilon}{16} \|\boldsymbol{q}(t)\| \|\boldsymbol{\delta}(t)\| \tag{13}
$$

and

$$
\frac{1}{2} \|\boldsymbol{\delta}(t)\| \leq \|\boldsymbol{d}(t)\| \tag{14}
$$

from (6). Noting

$$
\|\boldsymbol{q}(t+1)\|^2 - \|\boldsymbol{q}(t)\|^2 = 2\boldsymbol{q}(t)^T Q\boldsymbol{d}(t) + \|Qd(t)\|^2,
$$

we have

$$
\begin{aligned}
\|\boldsymbol{q}(t+1)\| - \|\boldsymbol{q}(t)\| &= \frac{\|\boldsymbol{q}(t+1)\|^2 - \|\boldsymbol{q}(t)\|^2}{\|\boldsymbol{q}(t+1)\| + \|\boldsymbol{q}(t)\|} \\
&= \frac{2\boldsymbol{q}(t)^T Q\boldsymbol{d}(t) + \|Qd(t)\|^2}{\|\boldsymbol{q}(t+1)\| + \|\boldsymbol{q}(t)\|} \\
&\leq \frac{2\boldsymbol{q}(t)^T \boldsymbol{d}(t) + \|Qd(t)\|^2}{\|\boldsymbol{q}(t)\|} \\
&= \frac{2\boldsymbol{q}(t)^T \left(\boldsymbol{\delta}(t) + (\boldsymbol{\beta}(t) - \mathbf{1}) \odot \boldsymbol{\delta}(t)\right) + \|Qd(t)\|^2}{\|\boldsymbol{q}(t)\|} \\
&= \frac{2\boldsymbol{q}(t)^T \boldsymbol{\delta}(t)}{\|\boldsymbol{q}(t)\|} + \frac{2\boldsymbol{q}(t)^T (\boldsymbol{\beta}(t) - \mathbf{1}) \odot \boldsymbol{\delta}(t)}{\|\boldsymbol{q}(t)\|} + \frac{\|Qd(t)\|^2}{\|\boldsymbol{q}(t)\|} \\
&\leq \frac{\varepsilon\|\boldsymbol{\delta}(t)\|}{8} + 2\|(\boldsymbol{\beta}(t) - \mathbf{1}) \odot \boldsymbol{\delta}(t)\| + \frac{\|d(t)\|^2}{R}
\end{aligned}
$$

Since

$$
\boldsymbol{\beta}(t) = (\beta_1(t), \cdots, \beta_p(t))^T \to \mathbf{1} \quad (t \to \infty)
$$

and

$$
\|\boldsymbol{d}(t)\| \to 0 \quad (t \to \infty),
$$

we can see that for sufficiently large $t$,

$$
\max_i |\beta_i(t) - 1| < \frac{\varepsilon}{8}
$$

and

$$
\|\boldsymbol{d}(t)\| < \frac{R\varepsilon}{8}.
$$

Now we have

$$
\|(\boldsymbol{\beta}(t) - \mathbf{1}) \odot \boldsymbol{\delta}(t)\| \leq \max_i |\beta_i(t) - 1|\|\boldsymbol{\delta}(t)\| \leq \frac{\varepsilon}{8}\|\boldsymbol{\delta}(t)\|,
$$

$$
\frac{\|d(t)\|^2}{R} \leq \frac{\varepsilon}{8}\|\boldsymbol{\delta}(t)\|.
$$

By (14), we obtain

$$
\begin{aligned}
&\|\boldsymbol{q}(t+1)\| - \|\boldsymbol{q}(t)\| \\
&\leq \frac{\varepsilon\|\boldsymbol{\delta}(t)\|}{8} + 2\|(\boldsymbol{\beta}(t) - \mathbf{1}) \odot \boldsymbol{\delta}(t)\| + \frac{\|d(t)\|^2}{R} \\
&\leq \frac{\varepsilon\|\boldsymbol{\delta}(t)\|}{8} + \frac{\varepsilon\|\boldsymbol{\delta}(t)\|}{4} + \frac{\varepsilon\|\boldsymbol{\delta}(t)\|}{8} \\
&= \frac{\varepsilon\|\boldsymbol{\delta}(t)\|}{2} \leq \varepsilon\|\boldsymbol{d}(t)\|.
\end{aligned}
$$

**Lemma 3.7.**

$$
\lim_{t \to \infty} \frac{\boldsymbol{v}(t)}{\|\boldsymbol{v}(t)\|} = \widehat{\boldsymbol{u}}. \tag{15}
$$

*Proof.* Since $\|\boldsymbol{d}(\tau)\| \to 0 \ (\tau \to \infty)$, we we can find a time $t_0$ such that for $\tau \geq t_0$,

$$
\|\boldsymbol{d}(\tau)\| \leq 1.
$$

By Lemma A.3 we can find a time $t_1 \geq t_0$ such that for $\tau \geq t_1$,

$$
\|\boldsymbol{d}(\tau)\| \leq \left(4\max_n \|\boldsymbol{\xi}_n\|\right) \|P\boldsymbol{d}(\tau)\|. \tag{16}
$$

Given $\varepsilon > 0$, by Lemma 3.6 we can choose $R \geq 1$ and $t_2 \geq t_1$ such that for $\tau \geq t_2$ and $\|\boldsymbol{q}(\tau)\| \geq R$,

$$
\|\boldsymbol{q}(\tau + 1)\| - \|\boldsymbol{q}(\tau)\| \leq \varepsilon\|\boldsymbol{d}(\tau)\|.
$$

Since $\|\boldsymbol{v}(\tau)\| \to \infty$ $(\tau \to \infty)$, we can choose $t_3 \geq t_2$ such that for $\tau \geq t_3$,

$$\|\boldsymbol{v}(\tau)\|^{-1} R < \varepsilon \tag{17}$$

and

$$\|\boldsymbol{v}(\tau)\|^{-1} \left( \|\boldsymbol{q}(t_2)\| + 4\,\varepsilon \max_n \|\boldsymbol{\xi}_n\| \|P\boldsymbol{v}(t_2)\| \right) < \varepsilon. \tag{18}$$

Now let $t \geq t_3$. To simplify notation we denote

$$\xi^* = \max_n \|\boldsymbol{\xi}_n\|.$$

**Case 1.** If $\|\boldsymbol{q}(t)\| < R$, then from (17) we directly obtain

$$\|\boldsymbol{v}(t)\|^{-1} \|\boldsymbol{q}(t)\| < \varepsilon. \tag{19}$$

**Case 2.** If for each $\tau \in \{t_2, \cdots, t\}$, $\|\boldsymbol{q}(\tau)\| \geq R$, then from (16),

$$
\begin{aligned}
\|\boldsymbol{q}(t)\| &= \|\boldsymbol{q}(t_2)\| + \sum_{\tau=t_2}^{t-1} \left( \|\boldsymbol{q}(\tau+1)\| - \|\boldsymbol{q}(\tau)\| \right) \\
&\leq \|\boldsymbol{q}(t_2)\| + \varepsilon \left( \|\boldsymbol{d}(t_2)\| + \cdots + \|\boldsymbol{d}(t-1)\| \right) \\
&\leq \|\boldsymbol{q}(t_2)\| + 4\,\varepsilon\,\xi^* \left( \|P\boldsymbol{d}(t_2)\| + \cdots + \|P\boldsymbol{d}(t-1)\| \right) \\
&= \|\boldsymbol{q}(t_2)\| + 4\,\varepsilon\,\xi^* \left( \|P\boldsymbol{d}(t_2) + \cdots + P\boldsymbol{d}(t-1)\| \right) \\
&= \|\boldsymbol{q}(t_2)\| + 4\,\varepsilon\,\xi^* \left\| P\left( \boldsymbol{d}(t_2) + \cdots + \boldsymbol{d}(t-1) \right) \right\| \\
&= \|\boldsymbol{q}(t_2)\| + 4\,\varepsilon\,\xi^* \left\| P\boldsymbol{v}(t) - P\boldsymbol{v}(t_2) \right\| \\
&\leq \|\boldsymbol{q}(t_2)\| + 4\,\varepsilon\,\xi^* \left( \|P\boldsymbol{v}(t_2)\| + \|P\boldsymbol{v}(t)\| \right).
\end{aligned}
$$

From (18) we have

$$
\begin{aligned}
\|\boldsymbol{v}(t)\|^{-1} \|\boldsymbol{q}(t)\| &\leq \|\boldsymbol{v}(t)\|^{-1} \left( \|\boldsymbol{q}(t_2)\| + 4\,\varepsilon\,\xi^* \|P\boldsymbol{v}(t_2)\| \right) + 4\,\varepsilon\,\xi^* \|\boldsymbol{v}(t)\|^{-1} \|P\boldsymbol{v}(t)\| \\
&< \varepsilon + 4\,\varepsilon\,\xi^* = (1 + 4\,\xi^*)\,\varepsilon.
\end{aligned} \tag{20}
$$

**Case 3.** If $\|\boldsymbol{q}(t)\| \geq R$ and there is a time $t_* \in \{t_2, \cdots, t-1\}$ such that

$$\|\boldsymbol{q}(t_*)\| < R,$$

then we can find the time $t^* \in \{t_*, \cdots, t-1\}$ such that

$$\|\boldsymbol{q}(t^*)\| < R$$

and for each $\tau \in \{t^* + 1, \cdots, t\}$,

$$\|\boldsymbol{q}(\tau)\| \geq R.$$

Thus we have

$$
\begin{aligned}
\|\boldsymbol{q}(t)\| &= \|\boldsymbol{q}(t^*)\| + \left( \|\boldsymbol{q}(t^* + 1)\| - \|\boldsymbol{q}(t^*)\| \right) + \sum_{\tau=t^*+1}^{t-1} \left( \|\boldsymbol{q}(\tau+1)\| - \|\boldsymbol{q}(\tau)\| \right) \\
&< R + \|Q\boldsymbol{d}(t^*)\| + \varepsilon \left( \|\boldsymbol{d}(t^* + 1)\| + \cdots + \|\boldsymbol{d}(t-1)\| \right) \\
&\leq R + \|\boldsymbol{d}(t^*)\| + 4\,\varepsilon\,\xi^* \left( \|P\boldsymbol{d}(t^* + 1)\| + \cdots + \|P\boldsymbol{d}(t-1)\| \right) \\
&\leq R + \|\boldsymbol{d}(t^*)\| + 4\,\varepsilon\,\xi^* \left( \|P\boldsymbol{d}(t_2)\| + \cdots + \|P\boldsymbol{d}(t-1)\| \right) \\
&= R + 1 + 4\,\varepsilon\,\xi^* \left( \|P\boldsymbol{d}(t_2) + \cdots + P\boldsymbol{d}(t-1)\| \right) \\
&= 2R + 4\,\varepsilon\,\xi^* \|P\boldsymbol{v}(t) - P\boldsymbol{v}(t_2)\| \\
&\leq 2R + 4\,\varepsilon\,\xi^* \left( \|P\boldsymbol{v}(t_2)\| + \|P\boldsymbol{v}(t)\| \right).
\end{aligned}
$$

Noting (17) and (18), we obtain

$$
\begin{aligned}
\|\boldsymbol{v}(t)\|^{-1} \|\boldsymbol{q}(t)\| &\leq \|\boldsymbol{v}(t)\|^{-1} \left( 2R + 4\,\varepsilon\,\xi^* \|P\boldsymbol{v}(t_2)\| \right) + 4\,\varepsilon\,\xi^* \|\boldsymbol{v}(t)\|^{-1} \|P\boldsymbol{v}(t)\| \\
&< 3\varepsilon + 4\,\varepsilon\,\xi^* = (3 + 4\,\xi^*)\,\varepsilon.
\end{aligned} \tag{21}
$$

Verifying (19), (20) and (21), we can see that, in any case, (21) is valid. Since $\varepsilon$ can be any positive number, we have

$$\lim_{t\to\infty} \frac{\|\boldsymbol{q}(t)\|}{\|\boldsymbol{v}(t)\|} = 0.$$

Thus

$$\lim_{t\to\infty} \frac{\|P\boldsymbol{v}(t)\|}{\|\boldsymbol{v}(t)\|} = 1, \quad \lim_{t\to\infty} \frac{\boldsymbol{q}(t)}{\|\boldsymbol{v}(t)\|} = \boldsymbol{0}.$$

Therefore,

$$\lim_{t\to\infty} \frac{\boldsymbol{v}(t)}{\|\boldsymbol{v}(t)\|} = \lim_{t\to\infty} \frac{P\boldsymbol{v}(t) + \boldsymbol{q}(t)}{\|\boldsymbol{v}(t)\|} = \lim_{t\to\infty} \frac{P\boldsymbol{v}(t)}{\|\boldsymbol{v}(t)\|} = \lim_{t\to\infty} \frac{\|P\boldsymbol{v}(t)\|}{\|\boldsymbol{v}(t)\|}\widehat{\boldsymbol{u}} = \widehat{\boldsymbol{u}}.$$

**Theorem 3.2.** AdaGrad iterates

$$\boldsymbol{w}(t+1) = \boldsymbol{w}(t) - \eta \boldsymbol{h}(t) \odot \boldsymbol{g}(t), \quad t = 0, 1, 2, \cdots, \tag{22}$$

has an asymptotic direction:

$$\lim_{t\to\infty} \frac{\boldsymbol{w}(t)}{\|\boldsymbol{w}(t)\|} = \frac{\widetilde{\boldsymbol{w}}}{\|\widetilde{\boldsymbol{w}}\|},$$

where

$$\widetilde{\boldsymbol{w}} = \operatorname*{arg\,min}_{\boldsymbol{w}^T\boldsymbol{x}_n \geq 1, \forall n} \left\|\boldsymbol{h}_\infty^{-1/2} \odot \boldsymbol{w}\right\|^2. \tag{23}$$

*Proof.* By hypothesis

$$\left\|\boldsymbol{h}_\infty^{1/2} \odot \boldsymbol{w}_\infty\right\|^2 = \min_{\boldsymbol{w}^T\boldsymbol{x}_n \geq 1, \forall n} \left\|\boldsymbol{h}_\infty^{1/2} \odot \boldsymbol{w}\right\|^2 = \min_{\left(\boldsymbol{h}_\infty^{1/2} \odot \boldsymbol{u}\right)^T \boldsymbol{x}_n \geq 1, \forall n} \|\boldsymbol{u}\|^2$$

$$= \min_{\boldsymbol{u}^T\left(\boldsymbol{h}_\infty^{1/2} \odot \boldsymbol{x}_n\right) \geq 1, \forall n} \|\boldsymbol{u}\|^2 = \min_{\boldsymbol{u}^T\boldsymbol{\xi}_n \geq 1, \forall n} \|\boldsymbol{u}\|^2.$$

Noting that both

$$\widehat{\boldsymbol{u}} = \operatorname*{arg\,min}_{\boldsymbol{u}^T\boldsymbol{\xi}_n \geq 1, \forall n} \|\boldsymbol{u}\|^2.$$

and $\boldsymbol{w}_\infty$ are unique, we must have $\widehat{\boldsymbol{u}} = \boldsymbol{h}_\infty^{-1/2} \odot \boldsymbol{w}_\infty$, or

$$\boldsymbol{w}_\infty = \boldsymbol{h}_\infty^{1/2} \odot \widehat{\boldsymbol{u}}.$$

From Lemma 3.7 and the relation

$$\boldsymbol{v}(t) = \boldsymbol{h}_\infty^{-1/2} \odot \boldsymbol{w}(t) \ (t = 0, 1, 2, \cdots),$$

we obtain

$$\boldsymbol{w}_\infty = \boldsymbol{h}_\infty^{1/2} \odot \lim_{t\to\infty} \frac{\boldsymbol{v}(t)}{\|\boldsymbol{v}(t)\|} = \lim_{t\to\infty} \frac{\boldsymbol{h}_\infty^{1/2} \odot \boldsymbol{v}(t)}{\|\boldsymbol{v}(t)\|} = \lim_{t\to\infty} \frac{\boldsymbol{w}(t)}{\|\boldsymbol{v}(t)\|}.$$

Thus

$$\lim_{t\to\infty} \frac{\boldsymbol{w}(t)}{\|\boldsymbol{w}(t)\|} = \lim_{t\to\infty} \frac{\|\boldsymbol{v}(t)\|}{\|\boldsymbol{w}(t)\|} \cdot \lim_{t\to\infty} \frac{\boldsymbol{w}(t)}{\|\boldsymbol{v}(t)\|} = \frac{\boldsymbol{w}_\infty}{\|\boldsymbol{w}_\infty\|}.$$

**Proposition 3.1.** Let $\boldsymbol{a} = (a_1, \cdots, a_p)^T$ be a vector satisfying $\boldsymbol{a}^T\boldsymbol{x}_n \geq 1$ $(n = 1, \cdots, N)$ and $a_1 \cdots a_p \neq 0$. Suppose that $\boldsymbol{w} = (w_1, \cdots, w_p)^T$ satisfies $\boldsymbol{w}^T\boldsymbol{x}_n \geq 1$ $(n = 1, \cdots, N)$ and

$$a_i (w_i - a_i) \geq 0 \ (i = 1, \cdots, p). \tag{24}$$

Then for any $\boldsymbol{b} = (b_1, \cdots, b_p)^T$ such that $b_1 \cdots b_p \neq 0$,

$$\operatorname*{arg\,min}_{\boldsymbol{w}^T\boldsymbol{x}_n \geq 1, \forall n} \|\boldsymbol{b} \odot \boldsymbol{w}\|^2 = \operatorname*{arg\,min}_{\boldsymbol{w}^T\boldsymbol{x}_n \geq 1, \forall n} \|\boldsymbol{w}\|^2 = \boldsymbol{a},$$

and therefore the asymptotic directions of AdaGrad (22) and GD

$$\boldsymbol{w}_G(t+1) = \boldsymbol{w}_G(t) - \eta \nabla \mathcal{L}\left(\boldsymbol{w}_G(t)\right) \quad (t = 0, 1, 2, \cdots), \tag{25}$$

are equal.

*Proof.* Without any loss of generality we may assume $a_i > 0 \ (i = 1, \cdots, p)$. Then (24) implies

$$w_i \geq a_i > 0 \ (i = 1, \cdots, p),$$

and thus

$$\|\boldsymbol{b} \odot \boldsymbol{w}\|^2 = b_1^2 w_1^2 + \cdots + b_p^2 w_p^2 \geq b_1^2 a_1^2 + \cdots + b_p^2 a_p^2.$$

Hence

$$\boldsymbol{a} = \arg\min_{\boldsymbol{w} \in F} \|\boldsymbol{b} \odot \boldsymbol{w}\|^2 = \arg\min_{\boldsymbol{w}^T \boldsymbol{x}_n \geq 1, \, \forall n} \|\boldsymbol{b} \odot \boldsymbol{w}\|^2.$$

By taking $\boldsymbol{b} = \boldsymbol{h}_\infty^{1/2}$, we get $\widetilde{\boldsymbol{w}} = \widehat{\boldsymbol{w}}$, where

$$\widehat{\boldsymbol{w}} = \arg\min_{\boldsymbol{w}^T \boldsymbol{x}_n \geq 1, \, \forall n} \|\boldsymbol{w}\|^2.$$

Thus the asymptotic direction of GD iterates (25), $\widehat{\boldsymbol{w}}/\|\widehat{\boldsymbol{w}}\|$, is equal to $\widetilde{\boldsymbol{w}}/\|\widetilde{\boldsymbol{w}}\|$, which is the asymptotic direction of AdaGrad iterates (22).

**Lemma A.6.** Suppose $N \geq p$ and the $p \times N-$matrix $\boldsymbol{X} = [\boldsymbol{x}_1, \cdots, \boldsymbol{x}_N]$, where

$$\boldsymbol{x}_n = (x_{n,1}, \cdots, x_{n,p})^T \quad (n = 1, \cdots, N),$$

satisfies the following conditions:

(i) For $n = 1, \cdots, p$,

$$x_{n,i} \begin{cases} > 0, & \text{for } i = n, \\ < 0, & \text{for } i \neq n. \end{cases}$$

(ii) The $p \times p-$matrix $\boldsymbol{X}_p = [\boldsymbol{x}_1, \cdots, \boldsymbol{x}_p]$ is nonsingular.

(iii) The unique solution $\boldsymbol{a} = (a_1, \cdots, a_p)^T$ of the linear system in $\boldsymbol{w}$

$$\boldsymbol{x}_n^T \boldsymbol{w} = 1 \quad (n = 1, \cdots, p). \tag{26}$$

satisfies $a_i > 0 \ (i = 1, \cdots, p)$.

Furthermore, suppose a vector $\boldsymbol{u} = (u_1, \cdots, u_p)^T$ satisfies

$$\boldsymbol{x}_n^T \boldsymbol{u} \geq 1 \quad (n = 1, \cdots, N), \tag{27}$$

then

$$u_i \geq a_i \quad (i = 1, \cdots, p). \tag{28}$$

*Proof.* For $n = 1$, we set

$$h_1 = \frac{1}{x_{1,1}} \left(\boldsymbol{x}_1^T \boldsymbol{u} - 1\right) \geq 0$$

and

$$\overline{u}_1 = u_1 - h_1 \leq u_1.$$

Denote $\boldsymbol{u}_1 = (\overline{u}_1, u_2, \cdots, u_p)^T$. Then

$$\boldsymbol{x}_1^T \boldsymbol{u}_1 = x_{1,1}\overline{u}_1 + x_{1,2}u_2 + \cdots + x_{1,n}u_n = \boldsymbol{x}_1^T \boldsymbol{u} - \left(\boldsymbol{x}_1^T \boldsymbol{u} - 1\right) = 1.$$

Since $x_{2,1} < 0$ and $\overline{u}_1 \leq u_1$, we have

$$\begin{aligned} \boldsymbol{x}_2^T \boldsymbol{u}_1 &= x_{2,1}\overline{u}_1 + x_{2,2}u_2 + \cdots + x_{2,n}u_n \\ &\geq x_{2,1}u_1 + x_{2,2}u_2 + \cdots + x_{2,n}u_n \\ &\geq \boldsymbol{x}_2^T \boldsymbol{u} \geq 1. \end{aligned}$$

Now we set

$$h_2 = \frac{1}{x_{2,2}} \left( \boldsymbol{x}_2^T \boldsymbol{u}_1 - 1 \right) \geq 0$$

and

$$\overline{u}_2 = u_2 - h_2 \leq u_2 \,.$$

Denote $\boldsymbol{u}_2 = (\overline{u}_1, \overline{u}_2, u_3, \cdots, u_p)^T$. Then

$$\boldsymbol{x}_2^T \boldsymbol{u}_2 = x_{2,1}\overline{u}_1 + x_{2,2}\overline{u}_2 + x_{2,3}u_3 + \cdots + x_{2,n}u_n$$
$$= \boldsymbol{x}_2^T \boldsymbol{u}_1 - \left( \boldsymbol{x}_2^T \boldsymbol{u}_1 - 1 \right) = 1.$$

Sequentially, we can define $\overline{u}_1, \cdots, \overline{u}_p$, such that

$$\overline{u}_n \leq u_n \quad (n = 1, \cdots, p). \tag{29}$$

Denote $\boldsymbol{u}_p = (\overline{u}_1, \cdots, \overline{u}_p)^T$. Then

$$\boldsymbol{x}_n^T \boldsymbol{u}_p = 1 \quad (n = 1, \cdots, p).$$

Noting that $\boldsymbol{a}$ is the unique solution of (26), we must have $\boldsymbol{u}_p = \boldsymbol{a}$, or

$$\overline{u}_n = a_n \quad (n = 1, \cdots, p),$$

which combined with (29) yields (28) .

**Proposition 3.2.** Suppose $N \geq p$ and $\boldsymbol{X} = [\boldsymbol{x}_1, \cdots, \boldsymbol{x}_N] \in \mathbb{R}^{p \times N}$ is sampled from any distribution whose density function is nonzero almost everywhere. Then with a positive probability the asymptotic directions of AdaGrad (22) and GD (25) are equal.

*Proof.* Let $\Pi_{n,i} : \mathbb{R}^{p \times N} \to \mathbb{R}$ be the projections defined as

$$\Pi_{n,i} (\boldsymbol{X}) = x_{n,i} \quad (n, \ i = 1, \cdots, p)$$

and let $G_{n,i} \subset \mathbb{R}^{p \times N}$ defined as

$$G_{n,n} = \{ \boldsymbol{X} \mid \Pi_{n,n} (\boldsymbol{X}) > 0 \}, \ \ G_{n,k} = \{ \boldsymbol{X} \mid \Pi_{n,k} (\boldsymbol{X}) < 0 \} \quad (n, \ k = 1, \cdots, p; \ \ k \neq n).$$

Then $G_{n,i}$'s are open, for all the projections are continuous.

Let $\boldsymbol{\Pi} : \mathbb{R}^{p \times N} \to \mathbb{R}^{p \times p}$ be the projection defined as

$$\boldsymbol{\Pi} (\boldsymbol{X}) = [\boldsymbol{x}_1, \cdots, \boldsymbol{x}_p]$$

and let $D \subset \mathbb{R}^{p \times N}$ defined as

$$D = \{ \boldsymbol{X} \mid \det (\boldsymbol{\Pi} (\boldsymbol{X})) \neq 0 \} \,.$$

Since both $\boldsymbol{\Pi}$ and $\det(\cdot)$ are continuous, $D$ is open.

Let $\Pi_i : \mathbb{R}^p \to \mathbb{R}$ be the projections defined as

$$\Pi_i (w_1, \cdots, w_p) = w_i \quad (i = 1, \cdots, p)$$

and let $A \subset \mathbb{R}^{p \times N}$ defined as

$$A = \left\{ \boldsymbol{X} \ \middle| \ \boldsymbol{X} \in D \,, \ \Pi_i \left( \left( \boldsymbol{\Pi} (\boldsymbol{X})^T \right)^{-1} \mathbf{1} \right) > 0 \ \ (i = 1, \cdots, p) \right\} \,,$$

where $\mathbf{1} = (1, \cdots, 1)^T \in \mathbb{R}^p$. Since all $\Pi_i$ are continuous, $A$ is open.

Clearly, for a matrix $\boldsymbol{X} \in \mathbb{R}^{p \times N}$, we have:

1) the condition (i) in Lemma A.6 holds, if and only if $\boldsymbol{X} \in G_{n,i}$ $(n, \ i = 1, \cdots, p)$;

2) the condition (ii) in Lemma A.6 holds, if and only if $\boldsymbol{X} \in D$;

3) the condition (iii) in Lemma A.6 holds, if and only if $\boldsymbol{X} \in A$.

Suppose $P$ is a distribution over $\mathbb{R}^{p \times N}$ and the density function of $P$ is nonzero almost everywhere. Let $\mathcal{S}$ be the set of all $p \times N-$matrices $\boldsymbol{X}$ satisfying conditions (i), (ii) and (iii). Then

$$\mathcal{S} = \left( \bigcap_{n,i=1}^{p} G_{n,i} \right) \bigcap D \bigcap A$$

is open in $\mathbb{R}^{p \times N}$. Thus $P(\mathcal{S}) > 0$.

For any $\boldsymbol{X} \in \mathcal{S}$, if $\boldsymbol{w} = (w_1, \cdots, w_p)$ satisfies $\boldsymbol{x}_n^T \boldsymbol{w} \geq 1 \ (n = 1, \cdots, N)$, then by Lemma A.6 we have

$$w_i \geq a_i \,, \text{ or } w_i - a_i \geq 0 \ (i = 1, \cdots, p) \,,$$

where $\boldsymbol{a} = (a_1, \cdots, a_p)^T$ is the unique solution of (26) with

$$a_i > 0 \ (i = 1, \cdots, p) \,.$$

Thus

$$\boldsymbol{w} = \boldsymbol{a} + (\boldsymbol{w} - \boldsymbol{a}) \in \left\{ \boldsymbol{a} + \boldsymbol{u} : \ \boldsymbol{u} = (u_1, \cdots, u_p)^T \text{ such that } a_i u_i \geq 0 \ (i = 1, \cdots, p) \right\} \,,$$

and the required conclusion follows from Proposition 3.1.