[Reviews · NeurIPS 2019]

Reviewer 1



The result is not that surprising, but to the best of my knowledge its statement is novel and it could be an interesting addition to the literature on implicit bias. It appears that the analysis mostly leverages the results of Soudry et. al. [2018], i.e. the authors make appropriate transformation to AdaGrad iterations and relate them to "induced" GD iterates. Also, event-hough simple, I find the small examples in Section 3.3.1 quite enlightening. On the other hand, not so convinced about the value of Prop. 3.1.(especially after seeing the stringent assumption of Lemma A.6)

Reviewer 2



The main contribution of the paper is to characterize the implicit bias of the adagrad on linear classification problems using logistic loss (and other losses). They show that the adagrad converges to a direction which is a solution of a quadratic optimization problem depending on the initial conditions, hyperparameters and the data itself unlike gradient descent which converges to the max margin direction. They also give a few toy examples to demonstrate the properties of the adagrad solution and the difference as compared to the gradient descent direction. Significance: It is an important problem to understand the implicit bias of various optimization algorithms on the solution and there have been several recent works along this direction. The motivation comes from understanding the generalization abilities of overparametrized deep neural networks. People have observed that the generalization abilities for deep neural networks with adagrad are not as good as with gradient descent and hence, understanding the exact form of implicit bias for adagrad is an interesting and important problem. Originality: This paper is the first to characterize the explicit direction to which adagrad converges for linear classification problems with logisitic or exponential loss. The framework and the assumptions and some parts of proofs are borrowed from Soudry et. al. However, overall the proof has a lot of new components as compared to Soudry et. al and they proceed via a different argument where they do not characterize the convergence rate just an estimate on the direction of convergence. [1] also talks about the convergence direction of adagrad depending on the initial conditions and the hyperparameters. Can the authors please discuss more on that in the related work section? Quality: Overall, I find the paper is well written. It would be nice to see more intuition of the proof of the convergence. A minor comment: 1) Appendix Page 4: line 43: \lambda is not defined and \eta seems to be missing here? [1] Gunasekar, Suriya, et al. "Characterizing implicit bias in terms of optimization geometry." ---------- I have read the authors' response and thank them for their response. It would have been nice to include some empirical results as a part of the paper on somewhat larger datasets than are currently in the feedback document.

Reviewer 3



Originality: The results are original to the best of my knowledge. Quality: Mostly looks OK. I only have an issue with Proposition 3.1. It is claimed that with positive probability property (i) of Lemma A.6 holds for any absolutely continuous distribution I don't understand why this is true. To the best of my understanding, such distributions are simply distributions with a density function. However, such distributions can have finite support (e.g., a uniform distribution over [0,1]), and therefore property (i) may have zero probability. Clarity: Mostly OK, but I feel the authors tend to focus on special cases, an don't give the complete picture, even when it's easy to explain/draw. For example, I was confused by the presentation of Example 3.1. It took me a while to understand the full solution: \tilde{w} is pointing in the (sign(cos(\theta),sin(\theta))) direction - except when \theta is pointing in the one of the axis directions, then the solution is non-unique. If the authors add (or draw) this general result, it will be much easier to understand what is going on in section 3.3.2. Significance: Personally, the results do not seem very surprising to me, but it is also important to prove results which are not surprising, and perhaps the proof method has some novelty. Minor comments: 1) line 81: "Lemma" -> "Theorem" 2) line 94: I guess it is assumed that y_n=0 w.l.o.g from this point on? Later y_n appears (e.g before line 146) again and should be noted why. 3) line 118: what is "decisive" part? 4) Example 3.1: Why use two points? Isn't one point enough? 5) line 151: what does "irrelevant to x_1" mean? %%% Edited after rebuttal %%% First, I thank the authors for clarifying Proposition 3.1, and I, therefore, increased my score as promised. The revised version seems correct now, but I think the authors overstate its significance in the rebuttal. First, it is not clear from the proposition, and the discussion around it, how small is this positive probability. The proof suggests that to me that "generically" this probability is exponentially vanishing with the number of dimensions and data points. Such a rarely occurring result does not seem to be very strong. Second, I think it is also somewhat inaccurate to say that "It negates the claim 'the implicit bias of AdaGrad does indeed depend on the initial conditions, including initialization and step size' on Page 8, Gunasekar, Suriya, et al. [2018a].". Specifically, I think the claim in Gunasekar et al. was not made for every dataset (as clearly, it not true with d=1), but that such dependency can exist, in contrast to gradient descent (and, indeed this seems to be the generic case from this paper). Still, I still think it is a borderline paper, as the final results are somewhat incremental (there are no surprises or new exciting theoretical directions), the characterization of the implicit bias seems incomplete (it still depends on h_{\infty}, which we don't know), and the writing still requires polishing (as admitted by the authors).

[Author Response · NeurIPS 2019]

We are grateful to all the reviewers, from whose critical reviews we have learned so much.

Prop. 3.1 explains that it is far beyond a rare exception that an AdaGrad scheme has an asymptotic direction independent
of the initial conditions. It negates the claim "the implicit bias of AdaGrad does indeed depend on the initial conditions,
including initialization and step size" on Page 8, Gunasekar, Suriya, et al. [2018a]. We think this is where the
significance of Prop. 3.1 lies in, though it is derived under certain stringent assumption as mentioned by Reviewer ♯1.

However, in the statement of Prop. 3.1, as pointed out by Reviewer ♯4, the term "absolutely continuous distribution"
was misused. It should be revised as "distribution whose density function is nonzero almost everywhere".

The arguments between Lemma 3.4 and Lemma 3.5 are neither a rigorous proof nor a sketch. They are devoted to
introducing certain objects and notations to be used in the rest of the paper, as well as some intuitions.

$\boldsymbol{\beta}(t) - \mathbf{1} \to \mathbf{0}$ is obvious, for $\boldsymbol{h}(t) \to \boldsymbol{h}_\infty$ and $\boldsymbol{\beta}(t) = \boldsymbol{h}_\infty^{-1} \odot \boldsymbol{h}(t) \to \mathbf{1}$ (see the formula above Line 93).

Some intuitions are provided in Line 94-98 and Line 106-124. We were very lucky to find that the *induced form* is
equivalent to the primary Adagrad scheme in studying the existence of the asymptotic direction. If one replaces $\boldsymbol{\beta}(t)$ in
the induced form with $\mathbf{1}$, the limit of $\boldsymbol{\beta}(t)$, then a GD scheme is obtained. Therefore we expected that the induced
form and the GD scheme have similar asymptotic behaviors. Thus the orthogonal decomposition in Line 116 emerges,
and we found that when the increment of the induced form is decomposed in this way, the projection in the asymptotic
direction of the GD scheme eventually becomes overwhelming. The accumulated effects determine the whole trend of
the induced form.

We are sorry to admit that there were a handful of minor errors in our paper. As noticed by Reviewer ♯4, the usage
of the symbol $y_n \in \{-1, 1\}$ might cause some confusion. In fact, from Line 75 to Line 140, we rewrote the loss
$\mathcal{L}(\boldsymbol{w}) = \sum_{n=1}^{N} l\left(y_n \boldsymbol{w}^T \boldsymbol{x}_n\right)$ as $\sum_{n=1}^{N} l\left(\boldsymbol{w}^T \boldsymbol{x}_n\right)$ by redefining $y_n \boldsymbol{x}_n$ as $\boldsymbol{x}_n$ to simplify notation. In Line 145 and
Example 3.1, we resumed the usage of $y_n$ to be in conformance with the setting of "separable data". Furthermore, as
pointed out by Reviewer ♯3, the formula in Line 43 on Page 4 of Appendix is inaccurate. It should be written as

$$\boldsymbol{\delta}(t)^T \widehat{\boldsymbol{u}} = -\eta \nabla \mathcal{L}_{ind}\left(\boldsymbol{v}(t)\right)^T \widehat{\boldsymbol{u}} = -\eta \sum_{n=1}^{N} l'\left(\boldsymbol{v}^T \boldsymbol{\xi}_n\right) \boldsymbol{\xi}_n^T \widehat{\boldsymbol{u}} \geq -\eta \sum_{n=1}^{N} l'\left(\boldsymbol{v}^T \boldsymbol{\xi}_n\right) > 0.$$

These errors mainly arose from the switching between different versions of the draft. Fortunately, they did not shake the
foundation of our reasoning at all.

It may be helpful to point out that our theory can be set up without the leverage of the results of Soudry et al. [2018].
In fact, our geometric estimation approach introduced in this paper can be used to give a much shorter proof for the
existence of the asymptotic direction of GD iterates, though without any convergence rate derived. Since our approach
does not rely on convergence rates, it may be applied to more cases where the learning rate $\eta$ is not a constant.

We did a few of numerical simulations to verify and illustrate our theory. We should have organized the paper better to
include some of the examples. Here is one of them.

Figure 1: **Left.** $\mathbf{x}_1 = \left(\cos\frac{\pi}{8}, \sin\frac{\pi}{8}\right)$ and $\mathbf{x}_2 = \left(\cos\frac{\pi}{20}, \sin\frac{\pi}{20}\right)$ are two support vectors. The red arrow denotes the direction of the max-margin separator. The blue and magenta arrows denote the asymptotic directions of AdaGrad iterates with $\eta = 0.1$ and $0.5$, respectively. The small angle between the two illustrates that the asymptotic direction may depend on $\eta$. However, all the asymptotic directions apparently diverge from the max-margin separator.

**Right.** The blue and green curves plot $\left\|\mathbf{w}(t)/\|\mathbf{w}(t)\| - \mathbf{d}_A\right\|$ vs. the number of the iterates with $\eta = 0.1$ and $0.5$, respectively, where $\mathbf{d}_A$ is the asymptotic direction of AdaGrad iterates. It can be observed that the directions of AdaGrad iterates slowly converge to $\mathbf{d}_A$. This aligns with Theorem 3.2.



[Meta-Review · NeurIPS 2019]

The paper examines the implicit bias of AdaGrad on linear classification with separable data. The authors show that using a sufficiently small step size, Adagrad converges in the direction of the minimal norm max-margin solution. The paper offers several interesting insights, including the novel convergence results for AdaGrad and interesting 2D examples. Overall, a good addition to the results focusing on understanding the implicit bias of optimization algorithms. Based on the feedback from the reviewers, the authors are encouraged to include some numerical results to confirm their theoretical findings and provide more proof details and novel ideas/insights in the main paper.